# Learning to Drive with Two Minds: A Competitive Dual-Policy Approach in Latent World Models

## Abstract

Recent advances in generative video models such as SORA have renewed interest in using world models to simulate physical dynamics for embodied decision-making tasks like autonomous driving. In parallel, end-to-end driving frameworks have begun to incorporate latent world models that predict future latent states as an auxiliary objective, trained jointly with imitation learning to enhance the model's planning capabilities. These models help encode environment dynamics and improve planning accuracy, but treat the world model as a passive auxiliary module. Separately, the Dreamer series has demonstrated the potential of using latent world models as simulators for reinforcement learning (RL), enabling agents to learn through imagined rollouts. However, combining imitation learning (IL) and RL in latent world models remains underexplored, and naive attempts to jointly optimize a shared policy often lead to instability and degraded performance. In this work, we propose a dual-policy framework that decouples IL and RL agents while sharing a common latent world model. The IL policy learns from expert driving data using supervised latent rollouts, while the RL policy explores the same latent environment via Dreamer-style training. Rather than fusing the two objectives, the agents are trained independently and compete during learning. Based on the outcome of their competition, knowledge—either expert behavior or exploratory experience—is selectively shared between agents. This architecture enables each policy to specialize while benefiting from the other's strengths. Experiments in complex driving scenarios demonstrate that our approach outperforms imitation-only baselines, leading to more robust and generalizable autonomous driving policies. We will release our code on GitHub soon.

## 1 Introduction

End-to-end (E2E) learning has emerged as a dominant architecture in autonomous driving [19, 21, 40], enabling direct optimization of the entire driving pipeline—including perception, prediction, and planning—toward the final driving objective. By bypassing modular decompositions, this paradigm mitigates issues such as error accumulation and misaligned optimization objectives.

Before the rise of E2E approaches, autonomous driving models typically relied on manually defined training objectives for each module. However, these objectives often introduced suboptimal biases when integrated into a full system. In contrast, E2E learning allows models to learn directly from expert demonstrations, focusing solely on observations and actions. Despite this advantage, E2E methods still struggle with the long-tail problem: expert-collected datasets rarely cover rare or safety-critical scenarios.

To address this, researchers have explored generative world models that conditionally synthesize driving observations based on actions or user-provided modalities (e.g. BEV and text) [38, 39].

These models can generate synthetic data to enrich training datasets, helping to alleviate long-tail issues. More recently, the combination of generative and reconstruction models has enabled high-fidelity scene generation [46, 33, 42], supporting closed-loop simulation and large-scale RL training [41]. However, these methods are computationally intensive, as they often require training a new reconstruction model (e.g., 3D Gaussian Splatting [23]) for each environment.

In this paper, we propose a more computationally efficient alternative for addressing the long-tail problem. Inspired by recent works integrating latent world models into end-to-end frameworks [28, 25], we adopt a latent world model rather than a pixel-level one. This design reduces computational overhead while preserving rich spatiotemporal representation. Our world model is trained jointly during the E2E imitation learning process, similar to model-based imitation learning (MBIL) [16]. The prediction objective enhances the model's ability to learn scene dynamics, aligning with predictive processing theories in human cognition [9].

While imitation learning allows agents to efficiently acquire expert behavior, it remains limited in generalization, especially in previously unseen scenarios—even with synthetic data augmentation. To overcome this, we draw inspiration from model-based reinforcement learning (MBRL), particularly the Dreamer series [12, 13, 14, 15]. We propose full utilize the latent world model with a learned reward model during E2E training rather than only set a new predictive task for the E2E model. This enables the agent to simulate imagined trajectories in latent space and optimize its policy using imagined rollouts.

By integrating model-based imitation and reinforcement learning into a unified framework, our agent benefits from both expert supervision and autonomous exploration. It can learn efficiently from demonstrations while improving through self-directed imagination-based learning in latent space.

Recent work such as RAD [6] also explores combining imitation and reinforcement learning. However, those methods rely on realistic 3D reconstructions for exploration, incurring significant computational cost. In contrast, our approach performs exploration entirely in latent space using a world model learned during imitation, eliminating the need for expensive 3D scene generation.

Our experimental results validate the effectiveness of our approach, achieving a substantial reduction in collision rate on the nuScenes benchmark, from 0.48% to 0.15%.

**Our contributions are summarized as follows:**

- We introduce a **competitive dual-policy architecture** that decouples imitation learning and reinforcement learning agents, enabling them to co-train within a shared latent world model while specializing in expert imitation and exploratory behavior, respectively.
- We propose a **compute-efficient RL framework for autonomous driving**, which eliminates the need for external simulators or costly scene reconstructions by leveraging a latent world model as an internal simulator for one step imagined.
- We conduct **extensive experiments on both the nuScenes and Navsim datasets**, showing that our approach outperforms imitation-only and prior world-model-based baselines, achieving state-of-the-art results in terms of both planning accuracy and safety.

## 2 Preliminary

In end-to-end autonomous driving, the model receives a sequence of past and current observations from multiple cameras mounted on the vehicle and predicts a future trajectory for ego car. At each timestamp $t$, the observation $o_t$ consists of multi-view images. Most end-to-end models can be viewed as encoder-decoder frameworks. They first encode the raw observations $o_t$ into intermediate features, often in the form of a bird's-eye view (BEV) representation $B_t$ using architectures like BEVFormer [43]. The BEV feature is then further processed to produce the latent scene state $s_t$:

$$s_t = \text{Encoder}(o_t), \tag{1}$$

After extracting scene state, the planning head directly predicts the future action sequence based on the latent state:

$$\hat{a}_{t:t+T_{plan}-1} = \text{PlanningHead}(s_t), \tag{2}$$

where $T_{plan}$ is the planning horizon, typically 3 to 4 seconds. Each action $a_t \in \mathbb{R}^2$ represents a 2D waypoint in the vehicle's local coordinate system, and the output sequence $\hat{a}_{t:t+T_{plan}-1}$ contains the predicted waypoints from $t$ to $t + T_{plan} - 1$.

The planning head is typically trained using expert demonstrations. The predicted action sequence is supervised using the corresponding ground-truth actions via an imitation loss:

$$L_{imi} = \sum_{\tau=0}^{T_{plan}-1} \|\hat{a}_{t+\tau} - a_{t+\tau}\|_1. \tag{3}$$

Recently, latent world models have been introduced into end-to-end autonomous driving to help the model better understand scene dynamics. Unlike pixel-level world models, latent world models operate in the abstract feature space, which makes them more computationally efficient. Given the current state $s_t$ and a proposed action sequence, the world model predicts a future state after $T_{fut}$ steps:

$$\hat{s}_{t+T_{fut}} = \text{LatentWorldModel}(s_t, \hat{a}_{t:t+T_{plan}-1}), \tag{4}$$

where the world model predict the state after $T_{fut}$ frames in the future, given current state $s_t$ and the proposed action sequence $\hat{a}_{t:t+T_{plan}-1}$.

To supervise the world model, we use the ground truth $s_{t+T_{fut}}$, which can be obtained by $o_{t+T_{fut}}$. For interpretability, we first convert the latent space state to BEV space, then supervise the world model's predictions in BEV space. This allow us to visualize the output of the latent world model directly:

$$B_{t+T_{fut}} = \text{BEVEncoder}(o_{t+T_{fut}}), \tag{5}$$

$$\hat{B}_{t+T_{fut}} = \text{Fusion}(\hat{s}_{t+T_{fut}}, B_t), \tag{6}$$

$$L_{wm} = \|\hat{B}_{t+T_{fut}} - B_{t+T_{fut}}\|_2. \tag{7}$$

## 3  Method

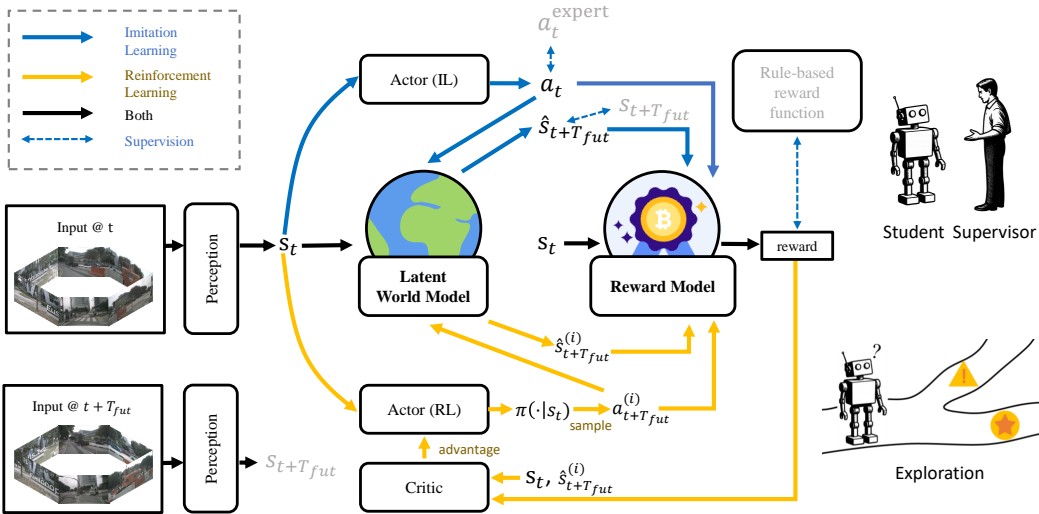

Figure 1: **Overview of Our Method.** We propose a dual-policy architecture that integrates imitation learning (IL) and reinforcement learning (RL) through a shared latent world model. During each iteration, both the IL Actor and RL Actor are trained simultaneously. The latent world model and reward model are learned during the IL phase. In the RL phase, only the RL Actor and Critic are updated. The RL Actor generates a policy from which $N$ action sequences are sampled. These sequences are evaluated using one-step imagination in the latent world model. Guided by the reward model and Critic, the RL Actor is optimized to assign higher probabilities to action sequences with higher estimated values.

## 3.1 Planning Head and Actor Modeling

Currently, most popular E2E autonomous driving model use a determined network to model the action sequence, showed in Eq.2. However, as the uncertainty and non-deterministic nature of planning [3] and the behaviors of the agents are highly multimodal [35], we follow this paradigm to build our planning head.

Firstly, since $s_t \in \mathbb{R}^{N_t \times d}$ contains almost all perception information, we use a set of waypoint querys $W_t \in \mathbb{R}^{N_c \times T_{plan} \times d}$ to extract the information the planning head needed for each action in the proposed action sequence:

$$s_t = \text{CrossAttn}(Q = W_t, K = s_t, V = s_t), \tag{8}$$

in which $N_t$ is the number of tokens to represent a scenario. Following the experiment of SSR [25], we set $N_t = 16$. And $d$ is the dimension of embedding, and $N_c$ is the number of high-level command (i.e. left turn, right turn, and go straight). Since autonomous driving vehicles usually require the navigation information to planning, most E2E datasets provide high-level commands for the model.

Then, intuitively, we should generate each action in the sequence step by step, adhering to causal relationships: first sampling $a_t \sim \pi(a_t|s_t)$, then $a_{t+1} \sim \pi(a_{t+1}|s_t, a_t)$, and so on, until $a_{t+T_{\text{plan}}-1} \sim \pi(a_{t+T_{\text{plan}}-1}|s_t, a_{t:t+T_{\text{plan}}-2})$.

However, we argue that humans often plan actions backward: they first identify the goal or endpoint and then reason back to generate the action sequence. In this case, we start from $a_{t+T_{\text{plan}}-1} \sim \pi(a_{t+T_{\text{plan}}-1}|s_t)$, then sample $a_{t+T_{\text{plan}}-2} \sim \pi(a_{t+T_{\text{plan}}-2}|s_t, a_{t+T_{\text{plan}}-1})$, and continue backward until $a_t \sim \pi(a_t|s_t, a_{t+1:t+T_{\text{plan}}-1})$.

To implement the invert chain rule, we adopt the masked attention mechanism with an invert attention mask, which is an inverted version of the causal mask in language model training. We first treat the scene token $s_t$ as a sequence of tokens with a horizon $T_{plan}$: $s_t = \{s_t^t, s_t^{t+1}, ..., s_t^{t+T_{plan}-1}\}$, where $s_t^{t+\tau} \in \mathbb{R}^{N_c \times d}$ is used to generate the $\tau$-th action of the proposed action sequence. The masked multi-head attention then enforces the inverted causal dependency during the generation process:

$$s_t = \text{MaskedMutiHeadSelfAttn}(Q = s_t, K = s_t, V = s_t, \text{mask}), \tag{9}$$

where mask is lower part of a triangular matrix, and $\text{mask} \in \mathbb{R}^{T_{plan} \times T_{plan}}$.

Here, we model each action in the action sequence with gaussian distribution:

$$\hat{a}_{t+\tau} \sim \pi(\hat{a}_{t+t}|s_t^{t+\tau}) = \mathcal{N}\left(\text{MLP}(s_t^{t+\tau}), \text{MLP}(s_t^{t+\tau})\right), \tag{10}$$

where two separate multi-layer perceptrons (MLPs) are used to generate the mean vector and covariance matrix of the Gaussian distribution respectively, for simplicity.

We update the imitation loss in Eq. 3 with a the negative log likelihood (NLL) loss as follows:

$$L_{imi} = \sum_{\tau=0}^{T_{plan}-1} \left\| E\left[\pi(\cdot|s_t^{t+\tau})\right] - a_{t+\tau} \right\|_1 - \mu \cdot \log \pi(a_{t+\tau}|s_t^{t+\tau}), \tag{11}$$

where $\mu$ is a hyper-parameter to balance the L1 loss and NLL loss.

## 3.2 Reward Modeling

Inspired by the cost module in LeCun's world model [24], which consists of two components—an intrinsic reward module that estimates the immediate "energy" of the current state (e.g., pain, pleasure, hunger), and a critic that predicts the expected cumulative future cost—we extend this design by introducing an additional imitation reward that evaluates the similarity between expert trajectories and those proposed by our model.

For the intrinsic reward, we focus solely on the current state to detect unsafe conditions, such as the ego vehicle being too close to surrounding dynamic agents or the road boundary. The critic reward then evaluates whether a proposed action sequence will lead to future collisions by considering the current state, the predicted future states, and the action sequence itself.

Beyond these rule-based components, we incorporate human driving preferences through an imitation reward, which measures the discrepancy between the model's proposed actions and those taken by human experts.

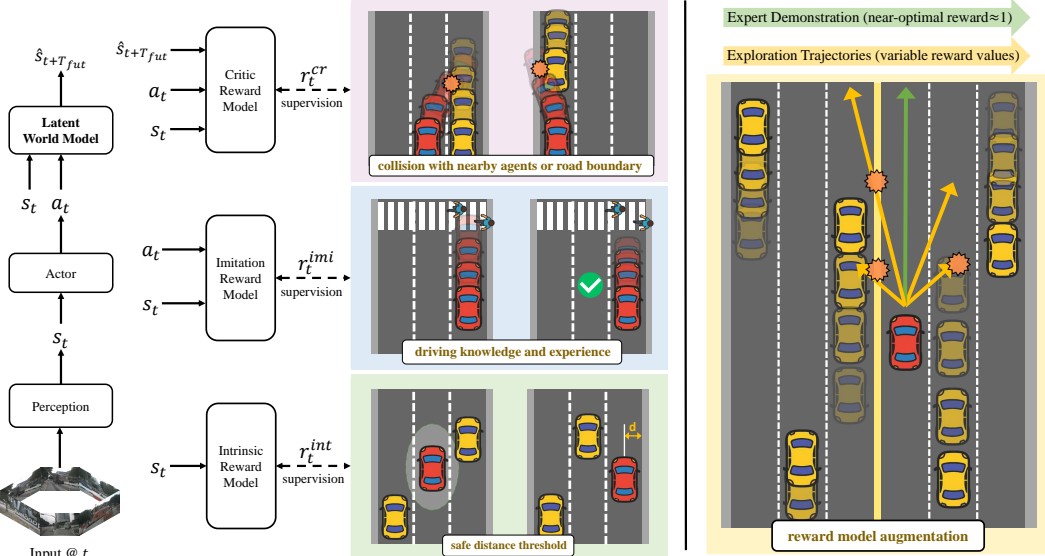

Figure 2: **Overview of Our Reward Model.** (Left) We design three separate reward models to provide intrinsic, critic-based, and imitation rewards. The intrinsic reward depends only on the current state, while the critic reward evaluates future states. The imitation reward measures the similarity between the predicted and expert action sequences. (Right) Since expert demonstrations typically yield near-optimal rewards, they provide limited learning signals for the reward model. To address this, we train the reward model using actions from a higher-variance (riskier) policy to encourage better reward shaping.

Thus, our reward model comprises three components (see Fig. 2):

$$
\begin{aligned}
\hat{r}_t^{int} &\sim p(\hat{r}_t^{int}|s_t), \\
\hat{r}_t^{imi} &\sim p(\hat{r}_t^{imi}|s_t, \hat{a}_{t:t+T_{plan}-1}), \\
\hat{r}_t^{cr} &\sim p(\hat{r}_t^{cr}|s_t, \hat{a}_{t:t+T_{plan}-1}, \hat{s}_{t+T_{fut}}).
\end{aligned}
\tag{12}
$$

We model each reward using a Gaussian distribution and compute ground-truth reward targets with rule-based heuristics. However, due to the open-loop nature of our prediction process, the computed rewards are often near-perfect, which limits the effectiveness of training. To address this, we introduce perturbations to the proposed action sequences, enabling the reward models to learn from both positive and negative samples.

We train the reward models using a negative log-likelihood loss:

$$
L_{\text{rew}} = -\log p(r_{int}|s_t) - \log p(r_{cr}|s_t, \hat{a}_{t:t+T_{plan}-1}, \hat{s}_{t+T_{fut}}) - \log p(r_{imi}|s_t, \hat{a}_{t:t+T_{plan}-1}). \tag{13}
$$

### 3.3 End-to-End Imitation Learning

Inspired by the world model training stage of Dreamer [15] and recent works that integrate world models into end-to-end autonomous driving [25, 28], we jointly train the driving policy, world model, and reward model by simply summing their individual losses (see Eq. 14).

$$
L_{e2e} = L_{imi} + L_{wm} + L_{rew}. \tag{14}
$$

### 3.4 Actor Critic Learning

Once the world model is trained via imitation learning, it can serve as a simulated environment for the reinforcement learning (RL) agent to explore. In this work, we adopt an actor-critic framework to optimize the actor module, where the critic is implemented as an MLP-based value estimator. However,

we observe that the differing optimization objectives of imitation learning (IL) and reinforcement learning can cause unstable training when both attempt to update the same actor. To address this, we decouple the IL and RL components by introducing a separate RL actor module, ensuring that each learning paradigm only updates its own actor.

The RL actor outputs a policy $\pi$, similar to IL actor (see Eq. 10). It also contains a critic head to estimate the return value distribution $v_t = E[p(\cdot|s_t)]$.We then sample $N$ future action sequence $a_{t:t+T_{fut}-1}^{(1)}, a_{t:t+T_{fut}-1}^{(2)}, ..., a_{t:t+T_{fut}-1}^{(N)}$. The world model acts as a one-step simulator and provides the interact feedback for the RL actor:

$$\hat{s}_{t+T_{fut}}^{(i)} = \text{WorldModel}(s_t, a_{t:t+T_{fut}-1}^{(i)}), \tag{15}$$

After the simulation, the critic estimates the value of the imagine state $v_{t+T_{fut}}^{(i)} = E[p(\cdot|s_{t+T_{fut}}^{(i)})]$, we use the expectation for the $\lambda$-return calculation (see Eq. 16) for each one step imagine chain and obtain $R_t^{(i)}$. We calculate the advantage value for each sample, where the reward $r_t = \hat{r}_t^{int} + \hat{r}_t^{cr} + \hat{r}_t^{imi}$:

$$R_t^{(i)} = r_t + \gamma \left((1-\lambda)v_t + \lambda R_{t+T_{fut}}^{(i)}\right), \quad R_{t+T_{fut}}^{(i)} = v_{t+T_{fut}}^{(i)}, \tag{16}$$

$$A^{(i)} = (R_t^{(i)} - v_t), \quad A^{(i)} = \frac{A^{(i)} - \text{mean}(A^{(1)}, A^{(2)}, ..., A^{(N)})}{\text{std}(A^{(1)}, A^{(2)}, ..., A^{(N)})}. \tag{17}$$

The computed advantages are used to update the RL actor via the actor loss. To further encourage exploration, we also add an entropy regularization term to the loss. For the critic, we use the average negative log-likelihood loss over all samples:

$$L_{\text{actor}} = -\frac{1}{N} \sum_{i=1}^{N} A^{(i)} \log \pi_{(t:t+T_{fut}-1)}^{(i)}|s_t), \quad L_{\text{critic}} = -\frac{1}{N} \sum_{i=1}^{N} \log v(R_t^{(i)}|s_t). \tag{18}$$

To enable knowledge sharing between the two actors—one trained via imitation and the other via exploration—we introduce a competitive mechanism that selectively updates their parameters, as described in Alg. 1.

---

**Algorithm 1** Competing IL and RL with Knowledge Blending

---

1: Initialize `actor_il`, `actor_rl`, `critic`
2: Initialize score_il $\leftarrow$ 0, score_rl $\leftarrow$ 0
3: **for** each $(\text{scene}, \texttt{expert\_act})$ in `dataset` **do**
4:     **if** `warmup` **then**
5:         `actor_il.learn_from_expert(scene, expert_act)`     ▷ Imitation learning (IL)
6:     **else**
7:         `actor_il.learn_from_expert(scene, expert_act)`
8:         `actor_rl.learn_from_reward(scene, critic)`   ▷ Reinforcement learning (RL)
9:         score_il += `actor_il.evaluate(scene, expert_act)`
10:        score_rl += `actor_rl.evaluate(scene, expert_act)`
11:       **if** `compete` **then**
12:         **if** score_il > score_rl **then**                ▷ soft update: rl ← r·il + (1–r)·rl
13:            `actor_rl ← blend(actor_il, actor_rl, r)`
14:         **else**                              ▷ soft update: il ← r·rl + (1–r)·il
15:            `actor_il ← blend(actor_rl, actor_il, r)`
16:         score_il ← 0, score_rl ← 0

---

# 4 Experiments

## 4.1 Datasets

**nuScenes [2]** is a large-scale autonomous driving benchmark featuring 1,000 20-second urban driving scenes with 1.4M annotated 3D boxes across 23 object classes. It provides 360° imagery from six

cameras and 2Hz keyframe annotations. Following prior works [19, 21], we evaluate planning using L2 placement error and Collision Rate.

**Navsim [5]** is a compact, filtered version of OpenScenes [4], itself derived from nuPlan [22]. It emphasizes challenging scenarios and contains 120 hours of driving at 2Hz. It includes *navtrain* and *navtest* splits for training and testing. To better reflect closed-loop safety and behavior, Navsim evaluates agents with six metrics: No at-fault Collisions (NC), Drivable Area Compliance (DAC), Time to Collision (TTC), Ego Progress (EP), Comfort (C), and Driving Direction Compliance (DDC). These are combined into a weighted Driving Score (PDMS).

### 4.2 Implementation Details

For experiments on nuScenes, our method builds upon the SSR framework [25]. We train our models using 8 NVIDIA A800-SXM4-80GB GPUs and perform evaluation on a single A800 GPU. The training is conducted with a batch size of 1 using the AdamW optimizer, with a learning rate set to $5 \times 10^{-5}$. All other training settings follow the original SSR configuration. The training process takes approximately 12 hours to complete.

For experiments on Navsim, we adopt the Transfuser [34] model as backbone. Transfuser employs a Transformer-based architecture to fuse front-view camera image and LIDAR data across multiple stages. We train our model on navtrain split and evaluate it on test split, using the same hardware configuration as in nuScenes experiments. The training is performed with a batch size of 16 using the AdamW optimizer, with a learning rare set to $1 \times 10^{-4}$.

### 4.3 Main Results

We evaluate our method on both the nuScenes [2] and Navsim [5] benchmarks. The results are presented in Tab. 1 and Tab. 2, respectively. For nuScenes, we follow the evaluation protocol of [21], which reports average L2 distance and collision rate over 1s, 2s, and 3s prediction horizons. For Navsim, we adopt the close-loop metrics provided in Navsim. Sepcifically, we use the test split rather than navtest split for evaluation, as the former contains much more scenarios (5044) than the later (885), making it more suitable for comprehensively assessing the model's overall driving performance.

Table 1: **Comparison of state-of-the-art methods on the nuScenes dataset.** Gray rows indicate methods that do not use additional supervision. *Ours and SSR are trained and evaluated on A800 GPUs, which differs from the original SSR paper. Results for other methods are reproduced from the SSR paper. Evaluation follows VAD metrics. The overall best results are highlighted in **bold**, while the best results among methods without additional supervision are underlined.

| Method | Auxiliary Task | L2 (m) ↓ | | | | Collision Rate (%) ↓ | | | |
|---|---|---|---|---|---|---|---|---|---|
| | | 1s | 2s | 3s | **Avg.** | 1s | 2s | 3s | **Avg.** |
| ST-P3 [18] | Det&Map&Depth | 1.33 | 2.11 | 2.90 | 2.11 | 0.23 | 0.62 | 1.27 | 0.71 |
| UniAD [19] | Det&Track&Map&Motion&Occ | 0.44 | 0.67 | 0.96 | 0.69 | 0.04 | 0.08 | 0.23 | 0.12 |
| VAD-Tiny [21] | Det&Map&Motion | 0.46 | 0.76 | 1.12 | 0.78 | 0.21 | 0.35 | 0.58 | 0.38 |
| VAD-Base [21] | Det&Map&Motion | 0.41 | 0.70 | 1.05 | 0.72 | 0.07 | 0.17 | 0.41 | 0.22 |
| BEV-Planner [31] | **None** | 0.28 | 0.42 | 0.68 | 0.46 | 0.04 | 0.37 | 1.07 | 0.49 |
| PARA-Drive [40] | Det&Track&Map&Motion&Occ | 0.25 | 0.46 | 0.74 | 0.48 | 0.14 | 0.23 | 0.39 | 0.25 |
| LAW [28] | **None** | 0.26 | 0.57 | 1.01 | 0.61 | 0.14 | 0.21 | 0.54 | 0.30 |
| GenAD [48] | Det&Map&Motion | 0.28 | 0.49 | 0.78 | 0.52 | 0.08 | 0.14 | 0.34 | 0.19 |
| SparseDrive | Det&Track&Map&Motion | 0.29 | 0.58 | 0.96 | 0.61 | 0.01 | 0.05 | 0.18 | 0.08 |
| UAD [10] | Det | 0.28 | 0.41 | 0.65 | 0.45 | 0.01 | 0.03 | 0.14 | **0.06** |
| SSR* [25] | **None** | 0.18 | 0.35 | 0.62 | **0.38** | 0.48 | 0.45 | 0.51 | 0.48 |
| Ours* | **None** | 0.21 | 0.40 | 0.69 | 0.43 | 0.09 | 0.11 | 0.23 | 0.15 |

On nuScenes, compared to the recent state-of-the-art SSR [25], which achieves the best L2 error (0.38), our method obtains a slightly higher L2 error (0.43), but significantly improves the collision rate. Ablation studies further confirm that each component of our approach contributes meaningfully to the final performance. Although UAD [10] achieves the lowest collision rate overall (0.06%), it

Table 2: **Comparison of state-of-art methods on Navsim test set.** *reproduced by us. † test on navtest set.

| Method | NC ↑ | DAC ↑ | TTC ↑ | Comf.↑ | EP ↑ | PDMS↑ |
|---|---|---|---|---|---|---|
| Human | 100.0 | 100.0 | 100.0 | 99.9 | 87.5 | 94.8 |
| Ego Status MLP | 93.0 | 77.3 | 83.6 | 100.0 | 62.8 | 65.6 |
| VADv2 [3] | 97.2 | 89.1 | 91.6 | 100.0 | 76.0 | 80.9 |
| UniAD [19] | 97.8 | 91.9 | 92.9 | 100.0 | 78.8 | 83.4 |
| PARA-Drive [40] | 97.9 | 92.4 | 93.0 | 99.8 | 79.3 | 84.0 |
| Transfuser [34] | 97.7 | 92.8 | 92.8 | 100.0 | 79.2 | 84.0 |
| LAW [28] | 96.5 | 95.4 | 88.7 | 99.9 | 81.7 | 84.6 |
| Hydra-MDP [30] | 98.3 | 96.0 | 94.6 | 100.0 | 78.7 | 86.5 |
| WoTE* [29] | 98.6 | 96.4 | 95.3 | 100.0 | 81.1 | 87.9 |
| DiffusionDrive† [32] | 98.2 | 96.2 | 94.7 | 100.0 | **82.2** | 88.1 |
| DiffusionDrive* [32] | 92.7 | 95.2 | 83.4 | 100.0 | 78.0 | 80.3 |
| Ours | **98.6** | **96.8** | **95.5** | **100.0** | 81.0 | **88.2** |

relies heavily on extensive supervision signals such as detection and tracking. In contrast, our method achieves the best collision rate (0.15%) among methods that do not use any auxiliary supervision beyond expert trajectories.

On navsim, our model obtains a PDMS of 88.2, outperforming recent state-of-art methods, showing notable improvements accross multiple sub-metrics, including NC (+0.4), DAC (+0.4) and TTC (+0.8). Compared to WoTE, which leverages a world model to evaluate candidate trajectories during testing, our approach achieves a higher overall score. While DiffusionDrive performs competitively on navtest split (88.1), its performance drops significantly on test split (80.3), indicating limited generalization to more diverse scenarios-possible due to the distribution shifts between navtest to test.

## 4.4 Ablation Study

We conduct an ablation study on the nuScenes dataset [2], evaluating the contribution of three key components: the inverse autoregressive module in the planning head, the reward model in end-to-end imitation learning, and the incorporation of reinforcement learning. Results are summarized in Tab. 3. The experiments demonstrate that each component contributes to improved performance across both planning accuracy and collision rate metrics. And we find that, the import of RL actor enhance the exploration, so that the L2 error becomes higher, but the agent learns to avoid colliding with others.

Table 3: **Ablation Study Results** on the impact of inverse autoregression (Inv. AR.), reward model (RM.), and reinforcement learning (RL). ↓: lower is better.

| Inv. AR. | RM. | RL | L2 (m) ↓ | | | | Collision Rate (%) ↓ | | | |
|---|---|---|---|---|---|---|---|---|---|---|
| | | | 1s | 2s | 3s | **Avg.** | 1s | 2s | 3s | **Avg.** |
| ✗ | ✗ | ✗ | 0.186 | 0.355 | 0.617 | 0.385 | 0.479 | 0.454 | 0.511 | 0.481 |
| ✓ | ✗ | ✗ | 0.200 | 0.377 | 0.652 | 0.410 | 0.098 | 0.166 | 0.329 | 0.198 |
| ✓ | ✓ | ✗ | **0.183** | **0.343** | **0.585** | **0.371** | **0.039** | 0.161 | 0.326 | 0.175 |
| ✓ | ✓ | ✓ | 0.206 | 0.399 | 0.694 | 0.43 | 0.098 | **0.112** | **0.228** | **0.146** |

## 4.5 Qualitative Results

Through qualitative analysis, we found that the latent BEV features effectively capture distinct driving behaviors. As shown in Fig. 3, due to the end-to-end nature of our model, the planning module directly influences the feature representations. For instance, in the "Turn Left" and "Turn Right" scenarios, the turning direction tends to exhibit higher activation (represented in red), indicating that these regions are more prominently perceived and attended to by the model. Regarding the

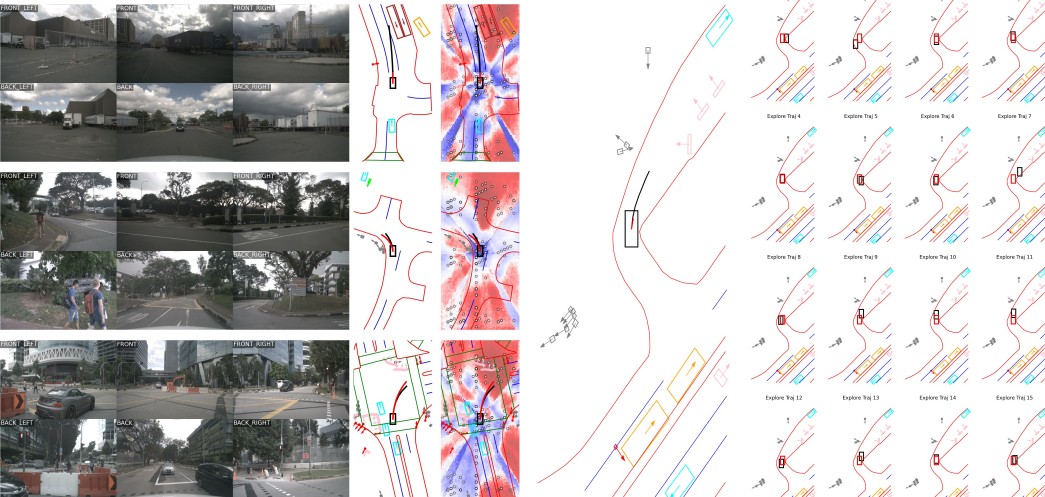

Figure 3: Qualitative visualization of model performance. The left column shows three driving scenarios (go straight, turn left, turn right), each with input camera views, ground-truth map, and an overlay of attention points (16 queries, 10 points/query) and latent BEV feature heatmaps. The right column shows closed-loop world model exploration, where red marks the ground-truth ego position and the black box the predicted one. We further analyze how the world model responds to different exploration trajectories by visualizing the resulting predicted perception features (see Fig. 4)

attention points, we observed that regardless of the driving context—whether going straight, turning left, or turning right—their spatial distribution remains relatively consistent. Interestingly, this pattern persists across all trained models. We attribute this phenomenon to the limitations of the nuScenes dataset, which may be insufficient for training a fully generalized policy, often leading to models that adopt fixed driving strategies. Moreover, we observed that when attention points are more dispersed across the BEV space and less concentrated near the ego vehicle, the L2 error tends to be lower. In contrast, when attention points are concentrated directly in front of and behind the vehicle, the collision rate is often reduced.

As illustrated on the right side of Fig. 3, the proposed method also demonstrates closed-loop behavior by leveraging the world model to simulate and explore diverse potential trajectories. These imagined scenarios are subsequently used to train the model, enabling it to acquire more generalized and robust driving strategies.

## 5   Conclusion

This work explores how to better utilize latent world models in recent world-model-based end-to-end autonomous driving (E2EAD) frameworks. We propose a novel approach that integrates the strengths of both imitation learning (IL) and reinforcement learning (RL) through a shared latent world model—without relying on external simulators. Our results show that latent world models can effectively serve as internal simulators for RL, significantly reducing collision rates and improving planning safety. Furthermore, visualization of the learned trajectories enhances the interpretability of using such latent dynamics for simulation. We believe our method offers new insights into leveraging world models for more robust and generalizable autonomous driving.

**Limitations and Future Work**: Despite its promising results, our approach has several limitations. Currently, we use relatively simple reward functions, which leads to a dominance of the IL policy in the competitive training framework. This is primarily because the IL agent benefits from dense and structured supervision, whereas the RL agent relies on sparse and delayed reward signals. In future work, we plan to explore more sophisticated reward designs and competition strategies to better balance learning between IL and RL agents. Additionally, we are interested in integrating the latent world model with more expressive generative models to bridge the gap between compact latent dynamics and explicit pixel-level understanding.

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

## A Related Work

### A.1 End-to-end Autonomous Driving

End-to-End automatic driving methods have recently made significant progress. These learning-based methods replaced traditional modular design by the end-to-end paradigm. [19] first demonstrated the potential of end-to-end by unifying perception and planning tasks within an unified framework. [21] vectorized the scene representation and improved the efficiency of inference. [40] decomposed the traditional end-to-end pipeline and searched for optimal architectures. some works tried to integrate world models into end-to-end automatic driving. [28] predicted future visual latens via world model, improving temporal understanding. [25] utilized sparse tokens to represent dense BEV features and similarly employed a world model to predict next Bird's Eye View (BEV) feature to enhance scene comprehension. [29] leverages a BEV-based world model to predict future agent states, enabling online trajectory evaluation and selection.

In contrast, our approach leverages the world model as an offline simulator for end-to-end automatic driving methods. Specifically, the AD model iteratively interacts with the world model to imagine future scene transitions through a world model, enabling reward-driven policy optimization in a simulated latent space.

### A.2 World Model

World model estimates the next state of the environment based on current state and action. It serves as a key component in both human and robot cognitive system to reason, predict and plan.[24]. The concept of world model was first introduced in [11], which designed a recurrent-style world model to predict future states, enabling the agent to learn an effective policy. Later, Yann Lecun proposed an agent architecture [24], including a world model to predict possible future world states given the input from perception, actor, and etc. Dreamer series [12, 13, 14, 15] advanced this idea by employing RSSM to model a world model from experience, helping to train a policy on imagined trajectories within latent space.

In autonomous driving, numerous world model-based methods have emerged. Video-based world models predict future driving videos conditioned on priors. [17] is a generative world model that utilizes various inputs to create realistic driving scenarios. [38] leveraged diffusion model to predict future images and planned based on these predictions. [7] was built on stable video diffusion [1] model and achieved high fidelity and versatile controllability. Occupancy-based world model like [47, 37, 44, 27] predict future occupancy condition on priors. [47] firstly defined the 4D occupancy forecasting task and used an auto-regressive framework to achieve spatial and temporal modeling. [37] further deployed diffusion model to model long-term temporal evolutions.

In contrast, our approach leverages world model in latent space, which greatly saves computational resources.

### A.3 Reinforcement Learning in Autonomous Driving

Reinforcement learning (RL) is an important technique for autonomous driving. [45] trained an RL expert to map BEV Input to actions, subsequently served as teacher for the end-to-end student model.

[20] leveraged a Vision-Language-Model (VLM) to generate rewards signals for RL. Some methods explicitly utilized a world model to simulate state-action transition between actor and environment to improve data efficiency, which is called Model-based reinforcement learning (MBRL). [26] integrated DreamV3 to train an expert driving model, becoming the fist agent to finish CARLA v2. [36] analyzed performance degradation sources of driving agents, proposing an adaptive fine-tuning strategy that selectively updates actor or world model. [8] proposed a novel framework by integrating Vista[7] with a multi-modal diffusion-based policy, achieving impressive performance in the CARLA.

In contrast, our fully E2E model conducts MBRL in the latent space and combines with IL policy to reach better performance.

## B Actor-Critic Preliminary

In model based RL algorithms, the world model is used to imagine future states based on proposed action sequence. More specifically, for each iteration, given an initial state, the world model and actor are invoked for $N_{img}$ imagine horizons. The actor proposes actions based on the state, and the world model predicts the future state based on these actions and the initial state, rolling out in this manner. For each rollout, the imagined state $s$, action $a$, and reward $r$ are stored for behavior learning.

There are two main components in the behavior learning process:

$$\text{Actor:} \quad \hat{a}_{t:t+T_{plan}-1} \sim \pi_\theta(\hat{a}_{t:t+T_{plan}-1}|s_t), \quad \text{Critic:} \quad v_\psi(R_t|s_t). \tag{19}$$

Here, $\pi_\theta$ represents the policy network, and the action sequence is sampled from the policy distribution. The value output of the critic is used to compute the return value, so here we use the expectation of the value distribution as its predicted value $v_t = E[v_\psi(\cdot|s_t)]$.

The return value is often computed using bootstrapped $\lambda$-returns to integrate the predicted rewards and values [15]:

$$R_T^\lambda = r_t + \gamma\left((1-\lambda)v_t + \lambda R_{t+T_{fut}}^\lambda\right), \quad R_{t+T_{fut}\cdot N_{img}}^\lambda = v_{t+T_{fut}\cdot N_{img}}, \tag{20}$$

where $r_t$ is the expectation value of reward distribution output by reward model $p_{\phi(r_t|s_t)}$ at timestamp $t$: $r_t = E[p_\phi(\cdot|s_t)]$. For the final roll out at timestamp $t + T_{fut} \cdot N_{img}$, the return value is equal to the reward value.

For the actor, the loss is composed of two parts: the reinforce term and an entropy regularizer to encourage exploration:

$$L_{actor} = -\sum_t \left[\underbrace{sg(R_t^\lambda - v_t)\cdot\log\pi(a_{t:t+T_{plan}-1}|s_t)}_{\text{reinforce}} + \underbrace{\eta\cdot H\left[\pi(a_{t:t+T_{plan}-1}|s_t)\right]}_{\text{entropy regularizer}}\right], \tag{21}$$

where the $sg(\cdot)$ denotes stop gradient.

For the critic, we use the negative log likelihood of the return in the value distribution as its loss:

$$L_{critic} = -\sum_t \log v(R_t^\lambda|s_t). \tag{22}$$

## C Additional Visualization

In this section, we show some additional visualization of our method.

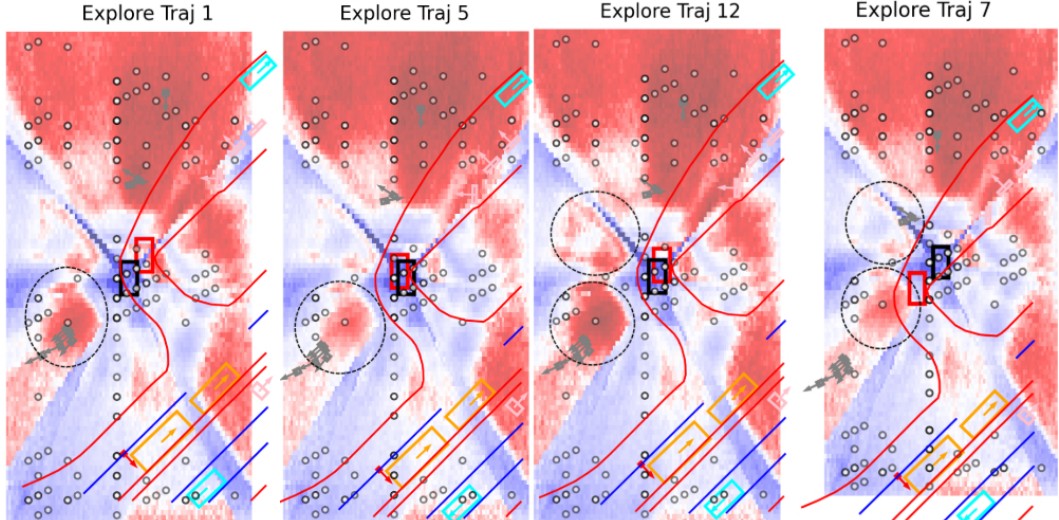

Figure 4: BEV features and attention point visualization during world model exploration, where red marks the ground-truth ego position and the black box the predicted one

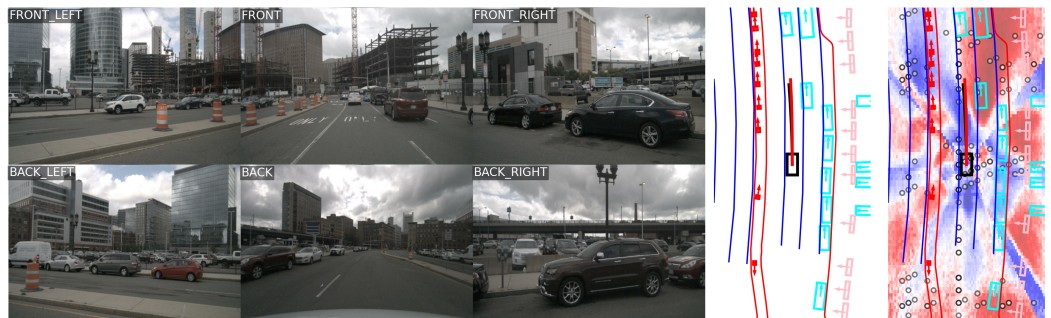

Figure 5: Additional qualitative results

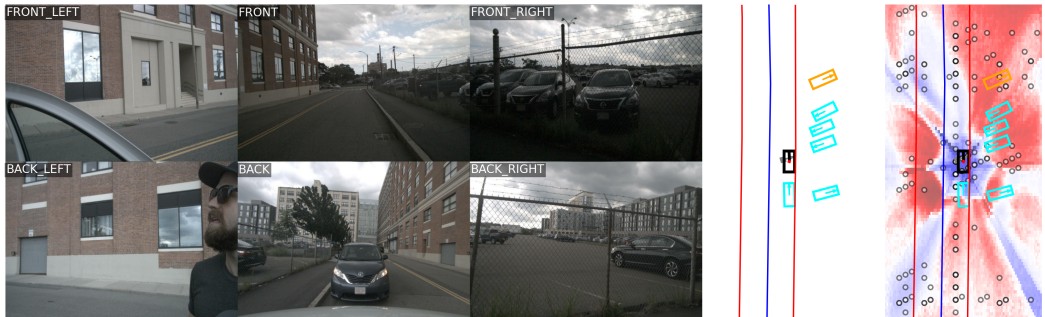

Figure 6: Additional qualitative results

