# OpenReview forum: "Learning to Drive with Two Minds: A Competitive Dual-Policy Approach in Latent World Models"
_NeurIPS.cc/2025/Conference — Submitted to NeurIPS 2025_

### Official Review · Reviewer_uMEo · 2025-07-02

**Clarity:** 1
**Significance:** 3
**Originality:** 2
**Rating:** 4
**Confidence:** 3

**Summary:**

This paper proposes a novel model-based approach to self-driving. This work takes inspiration from Dreamer and BEVFormer, and trains a world model in the BEV features space. The model is used for both imitation learning and RL, with two separate policies being trained at the same time. The authors propose a novel blending mechanism to let information flow between IL and RL policies. The authors demonstrate impressive results on nuScenes and navsim, particularly when comparing to other methods that do not use auxiliary tasks.

**Questions:**

- Q1 What's the reward $r_\mathrm{int}$? In the text, you say it's hardcoded. How exactly do you implement it? You say that your method doesn't use any auxiliary tasks, but in a way, depending on how this reward is implemented, you can consider this as an auxiliary task;
- Q2 In the conclusion, you say that the imitation policy often dominates the RL policy due to the nature of the data. Do you have a concrete number for how often RL beats IL throughout training and inference?
- Q3 How do you mix RL and IL policies during inference? Do you just use one of them? Or do you predict the outcome with the world model and use the actions you think are best according to the critic?

**Ethical Concerns:**

["NO or VERY MINOR ethics concerns only"]

**Final Justification:**

I enjoyed reading the paper, I think the direction of latent space dynamics models is promising for self-driving, but some parts were not clear, and the writing appeared sloppy. The authors addressed these issues, making me raise my score to 4. It's 4 and not higher because the paper is very specific to self-driving. Having more experiments with other domains would make the contribution more valuable.

**Limitations:**

yes

**Paper Formatting Concerns:**

no concerns

**Quality:**

3

**Strengths And Weaknesses:**

###### Strengths
- The proposed method does not use reconstruction loss, and operates fully in the latent space. This enables the representation to disregard irrelevant parts of the observation, such as e.g. foliage patterns in trees, and focus on the more relevant information. Additionally, this makes the method much more compute efficient.
- The proposed method achieves good results. In particular, collision rate of 15% on NuScenes is very good compared to other methods that don't use auxiliary tasks;
- Combining the knowledge of obtained from imitation learning and from RL is an exciting direction in the literature, and the idea proposed in this paper is quite interesting.

###### Weaknesses
- The writing is a bit confusing, some parts of the method are unclear (see questions).

I do think the idea is good, and the results are promising. I'm willing to raise my score if authors answer my questions and help me understand the method.

Typos and nitpicks:
52: full utilize -> to fully utilize
71: one step imagined ?
Figure 1: such large figures look better in the top part of the page
109: querys - queries
168: interact feedback -> interactive feedback
169: imagine state -> imagined state

These are not all, the paper has many typos, please run it through a spell checker.

---

> ### Author Rebuttal · Authors · 2025-07-31
>
> ## Meta Summary
>
> We sincerely thank all reviewers for their thoughtful feedback. Below, we summarize the core concerns raised across reviews.
> **We hope this summary helps the reviewers and area chair better understand the key review points and how we address them in our detailed responses:**
>
> 1. **Motivation and Justification for Dual-Policy Design**
>    Reviewers questioned the rationale behind combining IL and RL, and whether simpler alternatives (e.g., IL-pretrained world models for RL) would suffice. Several reviewers (yzWB, UrXC) requested ablations to isolate IL and RL contributions and clarify the necessity of the dual-policy framework.
>
> 2. **Clarity and Mechanism of Policy Blending**
>    Reviewers (yzWB, FwhP, uMEo) noted limited explanation of the competitive learning mechanism, including the blending method (soft vs. hard), threshold selection, and how arbitration occurs during training and inference.
>
> 3. **Effectiveness of Competition and Knowledge Transfer**
>    Concerns were raised about the empirical and theoretical support for the proposed competition mechanism. Reviewers asked for clearer evidence of its benefit and stability during learning.
>
> 4. **Motivation for Inverse Autoregressive Planning**
>    The rationale for planning actions in reverse (inverse autoregression) was seen as counterintuitive and insufficiently justified. Reviewers (UrXC, K5jx) requested both technical motivation and ablation results.
>
> 5. **Generalization and Long-Tail Evaluation**
>    Reviewers (K5jx, FwhP) emphasized the importance of testing the method in rare or challenging scenarios and questioned whether the hand-designed reward components can generalize to complex real-world driving situations.
>
>
> # Q1: Reward Design
> Thank you for raising this question. All of our reward models are learning-based neural networks that take different inputs and estimate predefined types of reward.
>
> Specifically, the intrinsic reward $r_{\text{int}}$​ you referred to is designed to assess the “comfort” or safety level of the current driving state. It takes the current state as input and predicts a score based on whether surrounding vehicles or pedestrians fall within a predefined safety distance from the ego vehicle.
>
> We agree that training these reward models involves an auxiliary task—namely, supervising the networks with rule-based scores computed from human-labeled data (e.g. the position of nearby agents). However, our approach differs from prior work that uses perception heads to predict dense supervision targets (e.g., 3D bounding boxes) to enhance perception.
>
> Instead, we follow a more lightweight and behavior-driven approach. Our reward models are not used to perceive every object in the scene; rather, they help the agent anticipate the consequences of its actions—e.g., whether a trajectory might result in a collision. This mimics how human drivers evaluate potential risks rather than reconstructing the full scene layout. In this sense, our auxiliary task is aligned more with outcome modeling than explicit perception.
>
> # Q2: Policy Usage
> Thank you for pointing this out. Below we provide the concrete cumulative win counts of the IL and RL actors throughout training, where a “win” means the policy achieved a higher reward.
>
> | Iteration | IL Wins | RL Wins | IL Win Rate |
> |-----------|---------|---------|-------------|
>  | 10k | 84 | 5 | 94.38% |
>  | 20k | 180 | 7 | 96.26% |
>  | 30k | 266 | 13 | 95.34% |
>  | 40k | 335 | 28 | 92.29% |
>  | 50k | 416 | 40 | 91.23% |
>  | 60k | 500 | 40 | 92.59% |
>  | 70k | 543 | 57 | 90.50% |
>  | 80k | 574 | 70 | 89.13% |
>
> As the table shows, the IL actor dominates the RL actor in most comparisons (with win rate consistently above 89%). In the early phase of training (before 60k iterations), imitation from expert demonstrations allows the IL actor to outperform RL consistently. However, as training progresses, the RL actor—leveraging knowledge gained from IL—begins to explore more effectively, leading to a gradual increase in its win rate.
>
> # Q3: Inference
> Thank you for this insightful question. In the current version of our paper, we only use the IL actor during inference. This decision was motivated by the following reasons:
>
> After competitive training, both the IL and RL actors share a large portion of their knowledge, since they co-evolve by learning from each other.
>
> Our current implementation models the RL actor as a sequence of six Gaussian distributions, each predicting a future waypoint. Directly sampling from these distributions can result in discontinuous or unrealistic trajectories (e.g., one point sampled far left, the next far right), leading to instability during deployment.
>
>
> Your suggestion led us to investigate this further. We designed an evaluation comparing the following inference strategies:
> - IL Actor only
> - RL Actor (using the mode of each Gaussian to generate trajectories deterministically)
> - Critic-Selected: IL provides 1 trajectory, RL samples 32; all are scored using the world model and value network, and the highest-scoring trajectory is chosen.
>
> | IL:RL | L2 (1s) | L2 (2s) | L2 (3s) | L2 (avg) | Col (1s) | Col (2s) | Col (3s) | Col (avg) |
> |-------------|---------|---------|---------|----------|----------|----------|----------|------------|
>  | IL | 0.3280 | 0.6506 | 1.0831 | 0.6872 | 0.098 | 0.171 | 0.531 | 0.267 |
>  | RL | 0.2950 | 0.5954 | 1.0105 | 0.6337 | 0.762 | 1.030 | 1.455 | 1.082 |
>  | Critic Select | 0.3297 | 0.6567 | 1.0923 | 0.6929 | 0.097 | 0.171 | 0.527 | 0.265 |
>
> As shown, the RL actor alone achieves slightly lower L2 error but exhibits significantly higher collision rates, suggesting that it generates less stable and less safe trajectories. This aligns with our observation that simple per-waypoint Gaussian modeling results in unrealistic outputs during inference.
>
> However, in the critic-selected setup, 97 out of 6019 test samples used a trajectory from the RL actor (i.e., scored higher than IL), and these helped reduce collisions in specific difficult cases—though the average metric shift is small. This also suggests that our learned reward model is effective at identifying safer trajectories, even from a diverse pool.
>
> To improve RL trajectory smoothness, we are currently exploring trajectory-level distribution modeling. One promising direction is to encode trajectories into a latent space using a VAE (e.g., 512-d latent vector), sample from a latent Gaussian, and decode a full trajectory. This helps ensure spatial and temporal continuity. We are also looking into diffusion-based trajectory generation methods (e.g., Diffusion-Planner), which we plan to integrate in future work.
>
> # Weaknesses
> Thanks for the comments.We correct those typos and nitpicks to enhance readability in our revised manuscript.
> ```
> 52: full utilize -> to fully utilize
> 53: rather than only set -> rather than only setting
> 59: explores combining -> explores the combination of
> 69: compute-efficient -> computation-efficient
> 71: one step imagined -> one-step imagination
> 94: the world model predict -> the world model predicts
> 98: This allow -> This allows
> 105: showed in Eq.2 -> as shown in Eq.2
> 109: querys-> queries
> 112: high-level command -> high-level commands
> 114: the navigation information to -> the navigation information for
> 127: is lower part of -> is the lower part of
> 167: N future action sequence -> N future action sequences
> 168: interact feedback -> interactive feedback
> 169: imagine state -> imagined state
> 170: one step imagine chain -> one-step imagined chain
> 202: learning rare -> learning rate
> 207: Sepcifically -> Specifically
> 209:  later (885) -> latter (885)
> 218: state-of-art -> state-of-the-act
> 219:  accoss multiple sub-metrics -> across multiple sub-metrics
> 229: And we find that, the import of RL actor enhance -> We find that the import of RL actor enhances
> 426: an unified -> a unified
> 428: some works -> Some works
> 431: next Bird's Eye View (BEV) -> the next Bird's Eye View (BEV)
> 440: both human and robot cognitive system -> both human and robot cognitive systems
>  The position of FIGURE has been changed to the top of the page and some minor punctuation and article errors have been corrected.
> ```

---

> > ### Comment · Reviewer_uMEo · 2025-08-02
> >
> > Thank you for answering my questions, this addresses most of my concerns. Please add these details to the paper.
> >
> > The experiment you show in Q3 is interesting, it's good to see that having the critic and world model choose between the proposed actions works pretty well. I wonder if this could potentially be combined with planning algorithms like MPPI [1]: you sample a few IL trajectories, a few RL trajectories, throw in a few random ones, then run a few steps of optimization to possibly find an even better action sequence.
> >
> > I will update my rating to 4.
> >
> > [1] Williams, Grady, Andrew Aldrich, and Evangelos Theodorou. "Model predictive path integral control using covariance variable importance sampling." arXiv preprint arXiv:1509.01149 (2015).

---

> > > ### Author Response · Authors · 2025-08-03
> > > **Thanks for Reviewer uMEo's valuable feedback**
> > >
> > > Thank you for the valuable suggestion — and also for taking the time to revisit your review and raise the score, we truly appreciate your thoughtful engagement!
> > >
> > > We find your proposal to integrate our current dual-policy selection mechanism with MPPI-style planning very inspiring. Currently, our method samples a few trajectories from both IL and RL actors and directly selects the best one based on the reward model's evaluation. As you pointed out, this process could benefit from iterative optimization, as in MPPI, to further refine the action sequence toward one that yields an even higher predicted reward.
> > >
> > > In essence, our approach already resembles a Mixture of Experts (MoE) architecture, where the IL actor specializes in expert-like behavior and the RL actor focuses on reward maximization and handling long-tail scenarios. Combining this setup with optimization techniques like MPPI could enhance robustness by exploiting both actor strengths more effectively.
> > >
> > > We agree this direction is promising and definitely worth exploring in future work. Of course, for real-time applications like autonomous driving, additional considerations such as inference latency and computational cost will also need to be addressed — but your insight provides a strong foundation.
> > >
> > > Thanks again for the constructive discussion and encouragement!

---

### Official Review · Reviewer_UrXC · 2025-07-03

**Clarity:** 2
**Significance:** 2
**Originality:** 3
**Rating:** 3
**Confidence:** 4

**Summary:**

The authors propose an approach towards training world models with both imitation learning and reinforcement learning in a stable manner. The approach leverages a reward model that models 3 types of rewards, competitive dynamics, as well as a policy that generates actions in reverse order. The results are promising on autonomous vehicle benchmarks.

**Questions:**

### Questions
- Line 98-99 "This allow us to visualize the output of the latent world model directly" --- how? I am confused about how the authors are able to visualize latents.
- Why does combining IL and RL become difficult in practice? Simply pretraining a world model with IL and then using it for the initialization of a world model in MBRL seems like a simple and stable baseline worth comparing to.
- I am still confused on why the proposed approach chooses to plan backwards. More motivation for this intuitively and from a technical perspective would help.

**Ethical Concerns:**

["NO or VERY MINOR ethics concerns only"]

**Final Justification:**

After reviewing the reviewer response and the discussions, some of my concerns have been addressed at a smaller scale with weaker experimentation. However, the many strange design decisions are still not justified, and as such I find it difficult to believe that the proposed approach will scale or have wide impact. Thus, I find it necessary the authors do one of the following:
A more thorough empirical investigation across many environments for the benefits of each of the proposed design decisions individually (and all of them in tandem)
Better motivation for the proposed design decisions (i.e. psychological evidence, or a strong intuitive explanation).

Although the authors did a great job of backing these decisions up with their single environment experiments, I find it very difficult to believe the proposed approach will scale or generalize to large-scale environments and tasks and therefore would like to see more evidence/motivation.

Update: After reviewing the newest rebuttal responses I feel better about the paper but am still inclined to reject as I still find the motivation and results weak for the proposed approach. I would say I'm between a borderline accept and borderline reject, but I maintain my perspective that a strong focus on motivation and strong empirical results on the many strange design decisions would make the proposed approach stand out. As of now, it seems mostly to be results that are niche and may not generalize well to new environments/tasks.

**Limitations:**

Yes

**Quality:**

2

**Strengths And Weaknesses:**

### Strengths
- The proposed approach achieved decent results in the nuScenes dataset and strong results on the Navsim test set.
- The proposed three separate reward model based approach seems intuitive, but ablations are needed to confirm this.
- The proposed approach is unique and different from existing approaches.

### Weaknesses
- I found the approach to be generally difficult to understand and that it did not make sense intuitively. I also found the proposed motivation somewhat difficult to follow. The design decision to model actions backwards was not well motivated and contradicts many current paradigms. In fact, the proposed Inv. AR approach worsens performance in L2 error.
- I would expect an ablation comparing the proposed approach to a more simple and standard approach where the world model is first pre-trained on the imitation learning data (similar to offline world modeling) and then used for MBRL as an initialized world model.
- It's unclear if the proposed competition aspect of the learning algorithm actually helped with learning improved world models. An ablation confirming whether this is the case would be helpful.
- The proposed competition based approach does not seem scalable, and although discussions in the limitations section discuss this, there are no experiments to confirm that this was even useful.
- Ultimately, the proposed approach seems promising but a lack of ablations for a huge amount of (uncommon) design decisions makes me question the success of the proposed approach.

### Other Comments
- The bolding in Table 2 in the Comf. column is a bit misleading.

---

> ### Author Rebuttal · Authors · 2025-07-31
>
> # W1: Hard to understand the proposed method
> Thank you for taking the time to review our paper. We apologize if parts of the method were unclear—we acknowledge that our writing could have been more accessible. At a high level, our goal is to integrate the strengths of both imitation learning (IL) and reinforcement learning (RL) into a latent world model-based end-to-end driving framework.
> Regarding your question on our inverse autoregressive (Inv. AR) planning strategy, we provide both motivation and supporting results in response to your Q3.
>
> # W2: Expect an ablation comparing with IL-pretrained + MBRL post training.
> Thanks for your suggestions, based on your suggestion and reviewer yZWB’s suggestion, we design an experiment to see whether or not our proposed two actor + competitive learning method is the best way to integrate IL and RL.
>
> We put the experiment table and analysis in Q2.
>
>
> # W3: Expect an ablation confirming whether the competition learning mechanism is useful.
> Thank you for the thoughtful suggestion. To clarify, the goal of our competitive learning mechanism is to enhance planning ability, not to improve the world model itself. The world model remains a lightweight simulator that supports the RL actor by enabling trajectory imagination and value estimation via the critic.
> To evaluate whether competition improves planning, we conducted an ablation study comparing models with and without the competitive training mechanism. In both cases, the IL and RL actors are trained with their respective objectives; the only difference is whether a competition-and-transfer mechanism is applied between them.
> | Experiment Description | Inference Policy | L2 (1s) | L2 (2s) | L2 (3s) | L2 (avg) | Col (1s) | Col (2s) | Col (3s) | Col (avg) |
> |------------------------|------------------|---------|---------|---------|----------|----------|----------|----------|-----------|
> | w/o competitive        | IL Actor         | 0.3215  | 0.6381  | 1.0705  | 0.6772   | 0.088    | 0.127    | 0.531    | 0.249     |
> | w/ competitive         | IL Actor         | 0.2953  | 0.6004  | 1.0252  | 0.6403   | 0.088    | 0.127    | 0.433    | 0.216     |
> | w/o competitive        | RL Actor         | 3.9218  | 6.5504  | 9.1841  | 6.5521   | 2.745    | 4.869    | 7.716    | 4.930     |
> | w/ competitive         | RL Actor         | 0.3110  | 0.6227  | 1.0442  | 0.6593   | 0.361    | 0.659    | 1.185    | 0.735     |
>
> First, we observe that the RL actor fails completely without competition, producing unstable and unsafe plans with very high L2 error and collision rates. This confirms that competitive learning—via selective knowledge transfer from the IL actor—is crucial for stabilizing RL learning in this setting.
> Second, we find that even the IL actor benefits from the presence of competition: its collision rate drops from 0.249% to 0.216%. This suggests that the RL actor's exploration provides useful feedback and improves robustness during IL training as well.
> These results support our claim that the competitive dual-policy framework enables mutual benefit between IL and RL, and helps both actors learn better planning behavior.
>
> # W4: About the scalability of your method.
> Thank you for raising this important point. Due to limitations in available datasets and compute resources, we were unable to provide large-scale experiments in this version. However, we can offer empirical evidence from our current experiments that shows the competitive learning dynamics between IL and RL policies throughout training.
> | Iteration | IL Wins | RL Wins | IL Win Rate |
> |-----------|---------|---------|-------------|
>  | 10k | 84 | 5 | 94.38% |
>  | 20k | 180 | 7 | 96.26% |
>  | 30k | 266 | 13 | 95.34% |
>  | 40k | 335 | 28 | 92.29% |
>  | 50k | 416 | 40 | 91.23% |
>  | 60k | 500 | 40 | 92.59% |
>  | 70k | 543 | 57 | 90.50% |
>  | 80k | 574 | 70 | 89.13% |
>
> These results suggest that while the IL actor dominates early training, thanks to dense supervision from expert demonstrations, the RL actor gradually improves by leveraging shared knowledge and exploring beyond the expert distribution. Even in this modest-scale setting, we observe an increasing win rate for the RL policy over time.
> We believe that scaling up the dataset will allow the IL actor to converge toward expert-level behavior, while the RL actor can continue exploring alternative strategies, potentially surpassing the expert’s performance in safety or efficiency. This reflects a key motivation behind our framework: IL captures expert behavior; RL pushes the boundary beyond it.
> As large-scale world models and reinforcement learning infrastructure (e.g., cloud-based training) gain adoption in industry, we anticipate that competitive IL+RL frameworks like ours will become increasingly relevant and impactful.
>
> # W5: Need more ablation
>
> Thank you for your suggestions, we have added three additional ablation study results:
> Methods to integrate IL and RL: Q2
> The necessity of competitive learning mechanism: W3
> Inverse autoregression usage: Q3
>
> # Q1: Visualizations
>
> Thank you for pointing this out. The reason we are able to visualize the latent outputs of the world model is as follows:
> 1. The latent output by the world model can be converted to a future BEV (bird’s eye view) feature $B \in R^{H \times W \times C}$ (see Eq.6 of our paper, Line 100).
> 2. These BEV features can be visualized in several ways: a) we can use PCA to reduce C to 3, mapping the result to an RGB image; b) Or by averaging the channel dimension to obtain a 2D matrix of shape [H, W], which can be rendered as a heatmap.
>
>
> # Q2: RL Post-training
>
> Thank you for this excellent question. In practice, combining IL and RL is not technically difficult, but the challenge lies in balancing their optimization objectives. Imitation learning (IL) aims to match expert behavior, while reinforcement learning (RL) seeks to maximize cumulative reward. These goals may align in some cases—but not always.
> Based on your suggestion and similar feedback from Reviewer yZWB, we conducted a comparative study of four integration strategies:
> - Loss-merging: Directly summing IL and RL losses to train a single actor.
> - IL-RL Interwave: Alternating training steps between IL and RL (experiment in progress).
> - Two-stage: Pretraining the model with IL, then fine-tuning it using only RL.
> - Decouple + Competitive (ours): Two separate actors trained via IL and RL respectively, exchanging knowledge through competition-based selection.
>
> The results are shown below:
>
> | Method                 | L2 (1s) | L2 (2s) | L2 (3s) | L2 (avg) | Col (1s) | Col (2s) | Col (3s) | Col (avg) |
> |------------------------|---------|---------|---------|----------|----------|----------|----------|-----------|
> | pure IL                | 0.3224  | 0.6274  | 1.0299  | 0.6599   | 0.088    | 0.122    | 0.456    | 0.222     |
> | loss-merging           | 0.3788  | 0.7250  | 1.1652  | 0.7563   | 0.029    | 0.117    | 0.540    | 0.229     |
> | IL-RL interwave        | 0.3181       | 0.6334      | 1.068       | 0.6733       | 0.117        | 0.171        | 0.537        | 0.275         |
> | two-stage              | 2.4303  | 4.2050  | 6.0318  | 4.2224   | 2.286    | 4.127    | 6.531    | 4.315     |
> | decouple + competitive | 0.2953  | 0.6004  | 1.0252  | 0.6403   | 0.088    | 0.127    | 0.433    | 0.216     |
>
>
> The loss-merging strategy underperforms because it mixes gradients from two objectives that are not always aligned. The two-stage approach, although usually used in LLM post-training, but here under our setting performs the worst. We believe this is due to distribution misalignment: the world model, having been pretrained solely on expert demonstrations via IL, fails to generalize to the novel state distributions encountered during RL-driven exploration.
> In contrast, our decouple + competitive approach achieves the best balance. It allows each policy to specialize and evolve independently while selectively exchanging knowledge through latent-space rollouts and reward-based selection. This setup avoids direct objective interference and produces more robust planning behavior, validating the motivation behind our dual-policy design.
>
> # Q3: Backward Planning
> In our method, we add an inverse autoregressive layer before decoding the trajectory to encourage the planning head to generate future waypoints from back on. This is inspired by how humans often plan: when aiming for a goal, we mentally project backward to adjust earlier steps.
>
> For example: Imagine a person sitting at a desk with a water bottle on the far right, partially blocked by an iPad. When the person wants to grab the bottle, they first attend to the goal (the bottle), realize that the iPad is in the way, and only then plan the specific motion (e.g., lift hand over the iPad).
>
> This goal-first reasoning aligns with backward planning, a concept also explored in embodied AI—e.g., "Efficient Robotic Policy Learning via Latent Space Backward Planning" (ICML 2025) adopts a similar strategy with improved efficiency and accuracy.
>
> Empirical Results: Before adopting Inv. AR, we conducted ablations comparing:
> - Baseline (no autoregression)
> - Baseline + Autoregression (AR)
> - Baseline + Inverse Autoregression (Inv. AR)
> The results are shown below:
>
> | Setting               | L2 (1s) | L2 (2s) | L2 (3s) | L2 (avg) | Col (1s) | Col (2s) | Col (3s) | Col (avg) |
> |-----------------------|--------|--------|--------|----------|----------|----------|----------|-----------|
> | Our model             | 0.2114 | 0.3932 | 0.6642 | 0.4227   | 0.1758  | 0.3272   | 0.4428   | 0.3153    |
> | Our model + AR        | 0.1943 | 0.3709 | 0.6429 | 0.4027   | 0.1660   | 0.2735   | 0.5600   | 0.3332    |
> | Our model + Inv. AR   | 0.1999 | 0.3773 | 0.6519 | 0.4097   | 0.09768   | 0.1660   | 0.3288   | 0.1975    |
>
> We found that after using the inverse autoregressive mechanism, L2 and Collision Rate got a better score. The experiment results push us to use Inv. AR in our model.

---

> ### Comment · Reviewer_UrXC · 2025-08-04
>
> Thanks for the comprehensive rebuttal, discussion, and additional experiments. I still believe many of the results obtained are specific to the environment used and are non-generalizable to other, more challenging domains. This makes it difficult to justify several of the non-standard design decisions made. I would have liked to see the proposed design decisions (inverse AR, competitive dynamics, triple resard) tested individually in many more environments to ensure they are helpful generally, and are not just overfitting to the specific environment (as no doubt they led to collision benefits in the discussed environment). For these reasons, I am maintaining my score.
>
> I also believe that doing inverse AR poses a fundamental challenge towards real-world adoption, as doing so with potentially online learning is impossible (as it relies on conditioning on future information).
>
> My recommendation would be to focus on very comprehensively showing each of the large design decisions (triple reward, inverse AR, competitive dynamics) each work very well on their own, and demonstrating these results hold on many environments. The proposed results seem to be the start of this, however, the lack of comprehensive experimentation on many benchmarks makes this challenging to trust.
>
> I also still find the evidence regarding planning backwards weak, and would have liked to see strong and broad psychological evidence for such a phenomenon or at least very general results that inverse AR helps more across environments/tasks.

---

> > ### Author Response · Authors · 2025-08-06
> > **Response to Reviewer UrXC's valuable feedback**
> >
> > Thank you for your detailed comments and suggestions.
> >
> > > no doubt they led to collision benefits in the discussed environment, but prefer evaluation of the proposed design decisions individually in many more environments
> >
> >
> >
> > In the rebuttal phase, we provide the ablation results of the competitive mechanism and dual policy under our settings (backbone:LAW, area: autonomous driving, dataset: nuScenes). And we are happy that our effort lets you be convinced of the result. For this paper, we will organize our work as you suggested, adding the ablation study of dual policy and competition in our manuscript separately.
> >
> >
> > > My recommendation would be to focus on very comprehensively showing each of the large design decisions (triple reward, inverse AR, competitive dynamics) each work very well on their own, and demonstrating these results hold on many environments.
> >
> > We completely agree that demonstrating generalizability is essential — thank you for emphasizing this important point.
> >
> > We also sincerely appreciate your suggestion to evaluate our design choices across a broader range of environments. In our current work, we have conducted experiments under the following settings:
> >
> > - Backbones: SSR (used in the main paper on nuScenes), LAW (used in the rebuttal for a stronger and more stable baseline on nuScenes), and WoTE (used in Navsim experiments)
> > - Datasets: nuScenes and Navsim
> > - Ablations: We have separately evaluated the dual-policy framework and competitive mechanism on nuScenes.
> >
> > We believe these evaluations offer useful insights for the autonomous driving community, especially regarding how each design component contributes under different backbones and datasets. That said, we fully acknowledge the importance of validating these components across diverse domains and tasks, and we plan to extend our experiments in that direction as part of future work.
> >
> > > doing inverse AR poses a fundamental challenge towards real-world adoption, as doing so with potentially online learning is impossible (as it relies on conditioning on future information)
> >
> > This is a thought-provoking point. However, we believe this may be a misunderstanding of how we apply inverse AR. In autonomous driving and many embodied tasks (e.g., manipulation), the agent is required to predict a full trajectory into the near future (e.g., 3 seconds ahead). The agent then executes only the first action from this predicted trajectory and replans at the next timestep.
> >
> > In our inverse AR setup, we do not condition on actual future information — instead, the trajectory is predicted in reverse order (from goal to start) using only current observations. The future context is imagined by the model (conditioned with the current observation), just as in standard forward models. So from an online deployment perspective, our method remains valid and feasible.
> >
> > > liked to see strong and broad psychological evidence for such a phenomenon or at least very general results that inverse AR helps more across environments/tasks
> >
> > Thank you for this observation. We agree that our paper did not provide enough support or motivation for inverse AR, and we will address this with both empirical and theoretical justifications in the revision.
> >
> > In addition to the accumulated error reduction argument (e.g., in AR, prediction errors can compound across future steps, whereas in inverse AR, the goal point can be predicted more reliably), we have found psychological studies supporting the benefits of backward planning:
> >
> > [1] Relative Effects of Forward and Backward Planning on Goal Pursuit (Psychological Science)
> >
> > [2] Backward Planning: Examining Consequences of Planning Direction for Time Prediction
> >
> > These studies show that backward planning improves performance, reduces bias, and enhances clarity in goal-related tasks — especially when tasks are complex. For example:
> >
> > > “Compared with forward planning, backward planning not only led to greater motivation, higher goal expectancy, and less time pressure but also resulted in better goal-relevant performance.”
> >
> > > “Backward planners were found to be less biased in their predictions... and generate more realistic predictions by influencing cognitive processes that normally lead to bias.”
> >
> > Recent experiments on tasks such as robot manipulation [3,4] also provide supporting evidence for this design choice.
> >
> > We will include these insights, along with ablation results, in the revised paper to provide clearer motivation and justification for our design.
> >
> > Thank you again for your thoughtful feedback. While we acknowledge that our work is an early step, we believe your suggestions will significantly strengthen its rigor and scope in future iterations.
> >
> > [3] Chain-of-Action: Trajectory Autoregressive Modeling for Robotic Manipulation
> >
> > [4] Efficient Robotic Policy Learning via Latent Space Backward Planning

---

> > > ### Author Response · Authors · 2025-08-09
> > > **Update the newest results**
> > >
> > > We’re excited to share our latest results:
> > >
> > > | Method     | L2 (1s) | L2 (2s) | L2 (3s) | L2 avg | Col% (1s) | Col% (2s) | Col% (3s) | Col% avg |
> > > |------------|---------|---------|---------|--------|-----------|-----------|-----------|----------|
> > > | LAW        | 0.3224  | 0.6274  | 1.0299  | 0.6599 | 0.088     | 0.122     | 0.456     | 0.222    |
> > > | Ours (LAW) | 0.3132  | 0.6105  | 1.0146  | 0.6461 | 0.039     | 0.078     | 0.479     | 0.199    |
> > >
> > > We sincerely thank you again for your time, constructive feedback, and guidance, your input has been instrumental in driving these improvements.

---

### Official Review · Reviewer_FwhP · 2025-07-03

**Clarity:** 3
**Significance:** 3
**Originality:** 3
**Rating:** 3
**Confidence:** 3

**Summary:**

This paper proposes a novel dual-policy framework for end-to-end autonomous driving that integrates imitation learning (IL) and reinforcement learning (RL) through a shared latent world model. Rather than fusing both learning objectives into a single actor, the authors introduce two independent policies—one trained via IL on expert rollouts and another via RL using imagined latent rollouts—and enable them to compete and share knowledge selectively. The world model is trained jointly with the IL actor and used as a one-step simulator for the RL actor. To support training, a hybrid reward model is designed, incorporating intrinsic safety constraints, imitation alignment, and critic-based value estimation. The proposed method achieves strong performance on nuScenes and the Navsim benchmark.

**Questions:**

1. How sensitive is the reward model to its designed components? Have you tested alternative reward formulations or considered using learned preference models?
2. Can the method handle complex scenarios? For example, how would the policy perform in unprotected left turns or with aggressive nearby agents?
3. Is there a risk of the RL policy exploring unsafe actions that the IL actor avoids?

**Ethical Concerns:**

["NO or VERY MINOR ethics concerns only"]

**Final Justification:**

After reading the last response and reviews from other reviewers, I think the contribution of RL actor is not very convincing, which leads to the doubt that if two minds if necessary. Many designs need more detailed explanation and verification. I look forward to a more solid version of this work. I recommend rejection.

**Limitations:**

yes

**Quality:**

3

**Strengths And Weaknesses:**

Strengths
1. The separation of IL and RL into distinct actors that compete and share parameters via performance-based blending is a novel design.
2. The use of a latent world model as a one-step internal simulator for RL avoids the need for expensive environment rollouts or high-fidelity 3D reconstructions.
3. The paper proposes a structured reward model combining intrinsic safety, imitation-based similarity, and future cost prediction. This multi-part reward design improves interpretability and supports safe exploration.
4. The method shows state-of-the-art or competitive performance on two benchmarks, outperforming several recent methods.

Weaknesses
1. Dual-policy training is creative but, it's unclear whether one actor may eventually dominate or if this setup guarantees consistent improvements.
2. Although the paper avoids simulator rollouts, the reward signal is composed of three manually designed components: an intrinsic safety score, imitation alignment, and critic-based feedback. Will these rule-based generalize across complex scenarios?
3. The paper proposes to blend policies based on performance comparisons but provides limited detail on how blending is executed (e.g., soft interpolation vs. hard switches), how the score thresholds are selected, and how stable this process is during training.
4. The RL actor operates in a latent world model with only one-step imagined rollouts. Will this shallow imagination restrict the agent’s ability to reason over long-term consequences?

---

> ### Author Rebuttal · Authors · 2025-07-31
>
> # W1: Policy usage
> We summarized the times of IL or RL wins in competition during training and inference. Below we provide the concrete cumulative win counts of the IL and RL actors throughout training, where a “win” means the policy achieved a higher reward.
> | Iteration | IL Wins | RL Wins | IL Win Rate |
> |-----------|---------|---------|-------------|
>  | 10k | 84 | 5 | 94.38% |
>  | 20k | 180 | 7 | 96.26% |
>  | 30k | 266 | 13 | 95.34% |
>  | 40k | 335 | 28 | 92.29% |
>  | 50k | 416 | 40 | 91.23% |
>  | 60k | 500 | 40 | 92.59% |
>  | 70k | 543 | 57 | 90.50% |
>  | 80k | 574 | 70 | 89.13% |
>
> As the table shows, the IL actor dominates the RL actor in most comparisons (with win rate consistently above 89%). In the early phase of training (before 60k iterations), imitation from expert demonstrations allows the IL actor to outperform RL consistently. However, as training progresses, the RL actor—leveraging knowledge gained from IL—begins to explore more effectively, leading to a gradual increase in its win rate.
>
> At this stage, the RL policy is modeled using a simple sequence of six Gaussian distributions. Despite this simplicity, the results already demonstrate the potential of our dual-policy mechanism. We believe that more sophisticated modeling of the RL policy in future work will lead to even stronger performance.
>
>
> # W2: Reward design
>
> We appreciate the reviewer’s concern. While our reward function is composed of three manually defined components: intrinsic safety, imitation alignment, and critic-based feedback, each was intentionally designed to be simple, lightweight, and general-purpose. Overly complex or domain-specific reward functions can negatively impact exploration efficiency and generalization in reinforcement learning.
> To assess whether these reward signals generalize well across scenarios, we evaluated the prediction accuracy of each reward model under both standard and long-tail conditions. All reward values are normalized to the range [0, 1], and we report L1 error between predicted and ground-truth reward. The longtail subsets were filtered from the eval set based on high L2 error or collision rate from a baseline model.
> | Reward Model | L1 Score (Eval Set) | L1 Score (Longtail_L2 Set) | L1 Score (Longtail_Col Set) |
> |--------------|----------------------|-----------------------------|------------------------------|
>  | Intrinsic | 0.0005 | 0.0005 | 0.0007 |
>  | Imitation | 0.1644 | 0.1755 | 0.1695 |
>  | Critic | 0.0609 | 0.0602 | 0.0879 |
>
>
>
> As expected, the intrinsic reward model yields extremely low L1 error, since expert drivers typically maintain safe distances—making the ground-truth score nearly always 1. While both the imitation and critic reward models show slightly higher L1 error on long-tail subsets, the increases are relatively small and remain within an acceptable range. This supports our claim that the selected reward signals are robust and learnable, even in challenging scenarios.
>
> # W3: Competitive mechanism detail
> Our policy blending strategy is based on periodic performance comparisons between the IL and RL actors, conducted every 100 training iterations. As outlined in Algorithm 1 of the paper, we define three blending behaviors based on the magnitude of their performance difference:
> Large difference (>10): We apply a hard switch, directly replacing the weights of the lower-performing actor with those of the better-performing one.
>
>
> Small difference (<1): We consider both actors to be performing similarly, and no update is made to either policy.
>
>
> Intermediate difference (1–10): We apply soft interpolation, blending the two policies by averaging their weights (i.e., the "loser" actor is updated to 50% of its current weights + 50% of the winner’s weights).
>
>
> As for how we selected the score thresholds (1 and 10), we first ran our baseline model and analyzed the distribution of planning scores over training. Based on this empirical analysis and early experiments, we chose these values to provide sufficient separation between clearly superior/inferior performance (hard swap), ambiguity (do nothing), and cases warranting partial knowledge sharing (soft interpolation). While this strategy was manually tuned, we found it to be stable and effective across training. The competition’s information during training is provided in W1.
>
> # W4: One-step imagination
> We agree that longer-horizon imagination may benefit long-term planning. However, we chose single-step imagination for training stability and computational efficiency, especially when using learned latent dynamics models.
>
> Despite its simplicity, we find that the combination of short-term latent rollouts and backward planning enables the model to make safe and foresighted decisions. Empirically, we observe clear gains in collision rate without sacrificing generalization. Additionally, recent work [1] LAW showed that the policy performs best when the world model predicts an intermediate horizon (1.5s) rather than longer time horizons (e.g. 3s).
>
> That said, our framework can be extended to multi-step rollouts in future work. We see this as a promising direction and plan to explore more expressive imagination horizons.
>
> # Q1: Reward head design
> Thank you for the insightful question.
> To assess the sensitivity of our system to the design of the reward model, we conducted an ablation comparing simple MLP-based reward models versus Transformer-based models. The results show that more expressive architectures (e.g., Transformers) consistently outperform simpler ones in capturing meaningful reward signals. Please see the table below for detailed results:
>
> | id                | L2 (1s) | L2 (2s) | L2 (3s) | Col (1s) | Col (2s) | Col (3s) |
> |-------------------|---------|---------|---------|----------|----------|----------|
> | MLP reward        | 0.2435  | 0.4569  | 0.7662  | 0.1954   | 0.3174   | 0.4883   |
> | Transformer reward| 0.2068  | 0.3944  | 0.6770  | 0.1954   | 0.2393   | 0.3614   |
>
>
>
> In this version, we focus on three core components: intrinsic safety, imitation alignment, and critic feedback. While effective, we agree that reward design in autonomous driving is a promising direction for future work. For example, in simulation platforms like NavSim, richer reward signals such as comfort or route smoothness could be incorporated. These could encourage smoother trajectories or more consistent speeds to improve passenger experience.
> Regarding learned preference models: prior works such as [1] have explored reinforcement learning from human feedback (RLHF) to fine-tune generative trajectory models. However, we argue that pure imitation—even from human preference data—tends to limit model performance to the level of human drivers. In contrast, our approach combines imitation and reinforcement to enable exploration and learning beyond the demonstrations, potentially surpassing human-level driving through optimization in the latent world model.
>
> [1] Finetuning Generative Trajectory Model with Reinforcement Learning from Human Feedback
>
> # Q2: Complex scenario
> We thank the reviewer for highlighting this important concern.
> To evaluate our method's robustness in challenging situations—such as unprotected left turns or interactions with aggressive nearby agents—we conducted additional experiments on a curated subset of the nuScenes dataset. This subset includes long-tail scenarios where the baseline LAW model exhibits high L2 errors or elevated collision rates.
> As shown in the tables below, our method consistently outperforms the baseline across both L2 and collision metrics in these complex scenarios. This suggests that our dual-policy framework, guided by human priors and refined through reinforcement, is more robust in long-tail edge cases commonly encountered in real-world driving.
>
> Long-tail Dataset Comparison (L2) Results
> | Model         | L2 (1s) | L2 (2s) | L2 (3s) | L2 (avg) | Col (1s) | Col (2s) | Col (3s) | Col (avg) |
> |---------------|---------|---------|---------|----------|-----------|-----------|-----------|------------|
> | LAW           | 0.3753  | 0.7388  | 1.2117  | 0.7753   | 0.0000    | 0.0795    | 0.4590    | 0.1795     |
> | LAW + ours     | 0.3172  | 0.6518  | 1.1143  | 0.6944   | 0.0000    | 0.0794    | 0.3531    | 0.1441     |
>
> Long-tail Dataset Comparison (Collision) Results
> | Model           | L2 (1s) | L2 (2s) | L2 (3s) | L2 (avg) | Col (1s) | Col (2s) | Col (3s) | Col (avg) |
> |-----------------|---------|---------|---------|----------|-----------|-----------|-----------|------------|
> | LAW             | 0.3360  | 0.6692  | 1.1837  | 0.7296   | 0.0000    | 0.8523    | 4.3561    | 1.7361     |
> | LAW + ours| 0.2996  | 0.6410  | 1.1126  | 0.6844   | 0.0000    | 0.2841    | 3.2197    | 1.1679     |
>
> These results demonstrate the potential of our approach to generalize beyond typical driving situations and improve safety and accuracy in more complex environments.

---

> > ### Comment · Reviewer_FwhP · 2025-08-05
> >
> > I appreciate the author's efforts on the rebuttal. I have below opinions:
> >
> > W1. I think there could be potential optimization space to balance the contribution of the two actors, since the IL wins at most of the time.
> >
> > W2. "Overly complex or domain-specific reward functions can negatively impact exploration efficiency and generalization in reinforcement learning. " Are there any empirical study for this claim?

---

> > > ### Author Response · Authors · 2025-08-06
> > > **Response to Reviewer FwhP's valuable feedback**
> > >
> > > > I think there could be potential optimization space to balance the contribution of the two actors, since IL wins most of the time.
> > >
> > > Thank you for pointing this out. In the current implementation, the RL actor generates trajectories by independently sampling six points from Gaussian distributions. This design leads to less smooth and less human-like trajectories, which likely explains why the IL actor dominates in most scenarios. To address this, we plan to improve the RL actor by first sampling trajectory features from a latent space and then decoding them into complete trajectories. We believe this will generate smoother and more realistic trajectories, enabling a more competitive dynamic between the two actors.
> > >
> > > > "Overly complex or domain-specific reward functions can negatively impact exploration efficiency and generalization in reinforcement learning. " Are there any empirical studies for this claim?
> > >
> > >
> > > Thank you for raising this important point. The concern about complex or overly specific reward functions harming exploration and generalization has been echoed in both theoretical discussions and empirical studies:
> > > - Nicholas et al.(2019) [1] emphasized that RL methods are often highly sensitive to reward design, which can be difficult to specify accurately in real-world domains.
> > > - Zhang et al. (2021) [2] highlighted that complex and sparse reward settings in urban driving make direct RL optimization challenging.
> > > - Li et al. (2024) [3] achieved promising performance using sparse and simple rewards to guide RL agents in realistic driving tasks.
> > >
> > > In line with these findings, we adopted a relatively simple reward structure in our work, using only three components: imitation alignment, critic-based score, and safety score. This design balances the need for effective learning signals while maintaining robustness and generalization.
> > >
> > > [1] Deep Imitative Models for Flexible Inference, Planning, and Control. 2019.
> > >
> > > [2] Roach: End-to-End Urban Driving by Imitating a Reinforcement Learning Coach. ICCV 2021.
> > >
> > > [3] Think2Drive: Efficient RL by Thinking with Latent World Model. ECCV 2024

---

### Official Review · Reviewer_yZWB · 2025-07-03

**Clarity:** 3
**Significance:** 2
**Originality:** 2
**Rating:** 3
**Confidence:** 4

**Summary:**

This paper introduces a _dual-policy framework_ for autonomous driving, where imitation learning (IL) and reinforcement learning (RL) agents are trained independently but share a latent world model. The IL policy leverages supervised latent rollouts from expert data, while the RL policy interacts with the same latent model via Dreamer-style training. Rather than combining IL and RL losses, the agents “compete” during training, with selective knowledge transfer—either expert-driven or exploration-based—depending on the outcome. This design aims to allow specialization while fostering mutual benefit. Experiments in complex driving scenarios show improved robustness and generalization compared to imitation-only baselines.

**Questions:**

1. **Deeper Ablation**: It would strengthen the paper to include a more detailed ablation of IL vs RL contributions. For instance, what happens when only IL is used with the latent model? Or when the RL policy receives no IL guidance?
2. **Policy Usage Statistics**: How often is each policy "trusted" during learning or inference? Quantitative statistics on policy arbitration would help clarify how the knowledge transfer actually happens in practice.

**Ethical Concerns:**

["NO or VERY MINOR ethics concerns only"]

**Final Justification:**

While the proposed dual-policy framework is an interesting integration of IL and RL, I remain unconvinced of its broader utility. The improvements over IL-only baselines are small, and the evidence for cross-domain generalization is limited. More concerning, the authors replaced the original SSR backbone with LAW during the rebuttal phase due to reproducibility issues. Changing a core component mid-review not only raises fairness concerns but also highlights the limited scope and stability of the experiments. Given these issues, along with the lack of statistical significance testing, my recommendation remains unchanged.

**Limitations:**

Yes

**Quality:**

3

**Strengths And Weaknesses:**

**Strengths:**

- **Latent World Model Usage**: The method avoids costly external simulators or scene reconstructions by using a latent model for imagination-based training. This is a compelling and compute-efficient design, particularly relevant to real-world autonomous driving settings.
- **Modular Dual-Policy Setup**: The separation of IL and RL agents is conceptually clean, and the mutual competition is a novel touch.
- **Empirical Results**: The experiments are convincing and suggest a tangible benefit over baseline methods.

**Weaknesses:**
- **Scope of Contribution**: The algorithmic novelty may be limited for a NeurIPS-style audience; the work feels more suitable for a robotics or autonomous driving venue due to its system-level focus rather than core RL or representation learning advancements.
- **Evaluation of Core Mechanism**: The proposed dual-policy competition mechanism is underexplored. There is little analysis or ablation of how the competition dynamics impact learning, and no theoretical or empirical justification for why this form of policy interaction should work better than fused or alternating training.
- **Lack of Theoretical Insight**: No formal analysis or grounding is provided for the knowledge sharing or competitive mechanism. The method remains empirical and heuristic-driven.

---

> ### Author Rebuttal · Authors · 2025-07-31
>
> # Meta Summary
> We sincerely thank all reviewers for their thoughtful feedback. Below, we summarize the core concerns raised across reviews. We hope this summary helps the reviewers and area chair better understand the key review points and how we address them in our detailed responses:
>
> Motivation and Justification for Dual-Policy Design
> Reviewers questioned the rationale behind combining IL and RL, and whether simpler alternatives (e.g., IL-pretrained world models for RL) would suffice. Several reviewers (yzWB, UrXC) requested ablations to isolate IL and RL contributions and clarify the necessity of the dual-policy framework.
>
> Clarity and Mechanism of Policy Blending
> Reviewers (yzWB, FwhP, uMEo) noted limited explanation of the competitive learning mechanism, including the blending method (soft vs. hard), threshold selection, and how arbitration occurs during training and inference.
>
> Effectiveness of Competition and Knowledge Transfer
> Concerns were raised about the empirical and theoretical support for the proposed competition mechanism. Reviewers asked for clearer evidence of its benefit and stability during learning.
>
> Motivation for Inverse Autoregressive Planning
> The rationale for planning actions in reverse (inverse autoregression) was seen as counterintuitive and insufficiently justified. Reviewers (UrXC, K5jx) requested both technical motivation and ablation results.
>
> Generalization and Long-Tail Evaluation
> Reviewers (K5jx, FwhP) emphasized the importance of testing the method in rare or challenging scenarios and questioned whether the hand-designed reward components can generalize to complex real-world driving situations.
>
> # W1: NeurIPS
> We thank the reviewer for raising the point. While our work may appear system-level design at first glance, the structure of the method is not a simple pipeline or incremental engineering but rather an innovative design to address the problems in the field of autonomous driving. We believe our paper is of interest to the broader NeurIPS audience.
>
> # W2: Lack Evaluation of Core Mechanism
>
> Thanks for your suggestion!
>
> ## 2.1 Ablation of how the competition dynamics impact learning
> To evaluate whether competition improves planning, we conducted an ablation study comparing models with and without the competitive training mechanism. In both cases, the IL and RL actors are trained with their respective objectives; the only difference is whether a competition-and-transfer mechanism is applied between them.
>
> | Experiment Description | Inference Policy | L2 (1s) | L2 (2s) | L2 (3s) | L2 (avg) | Col (1s) | Col (2s) | Col (3s) | Col (avg) |
> |------------------------|------------------|---------|---------|---------|----------|----------|----------|----------|-----------|
> | w/o competitive        | IL Actor         | 0.3215  | 0.6381  | 1.0705  | 0.6772   | 0.088    | 0.127    | 0.531    | 0.249     |
> | w/ competitive         | IL Actor         | 0.2953  | 0.6004  | 1.0252  | 0.6403   | 0.088    | 0.127    | 0.433    | 0.216     |
> | w/o competitive        | RL Actor         | 3.9218  | 6.5504  | 9.1841  | 6.5521   | 2.745    | 4.869    | 7.716    | 4.930     |
> | w/ competitive         | RL Actor         | 0.3110  | 0.6227  | 1.0442  | 0.6593   | 0.361    | 0.659    | 1.185    | 0.735     |
>
> First, we observe that the RL actor fails completely without competition, producing unstable and unsafe plans with very high L2 error and collision rates. This confirms that competitive learning—via selective knowledge transfer from the IL actor—is crucial for stabilizing RL learning in this setting.
>
> Second, we find that even the IL actor benefits from the presence of competition: its collision rate drops from 0.249% to 0.216%. This suggests that the RL actor's exploration provides useful feedback and improves robustness during IL training as well.
> These results support our claim that the competitive dual-policy framework enables mutual benefit between IL and RL, and helps both actors learn better planning behavior.
>
> We also provide the specific number of IL Actor wins and RL Actor wins in the training stage in Q2.
>
> ## 2.2 Why this form of policy interaction should work better than fused or alternating training.
>
> Thanks for your question, we will provide an ablation study at Q1.
>
> # W3
> Regarding theoretical analysis, we acknowledge that our current submission is primarily empirical. However, the mechanism of combining IL and RL has not been fully explored by previous works. Our results demonstrate that these designs lead to performance improvements. We see formal analysis of IL-RL competition as a valuable direction for future work and plan to pursue this further.
>
> # Q1: deeper ablation
> Based on your suggestion and similar feedback from Reviewer UrXC, we conducted a comparative study of four integration strategies:
> - Loss-merging: Directly summing IL and RL losses to train a single actor.
> - IL-RL Interwave: Alternating training steps between IL and RL.
> - Two-stage: Pretraining the model with IL, then fine-tuning it using only RL.
>
>
> Decouple + Competitive (ours): Two separate actors trained via IL and RL respectively, exchanging knowledge through competition-based selection.
>
> The results are shown below:
>
> | Method                 | L2 (1s) | L2 (2s) | L2 (3s) | L2 (avg) | Col (1s) | Col (2s) | Col (3s) | Col (avg) |
> |------------------------|---------|---------|---------|----------|----------|----------|----------|-----------|
> | pure IL                | 0.3224  | 0.6274  | 1.0299  | 0.6599   | 0.088    | 0.122    | 0.456    | 0.222     |
> | pure RL                | 3.9218  | 6.5504  | 9.1841  | 6.5521   | 2.745    | 4.869    | 7.716    | 4.930     |
> | loss-merging           | 0.3788  | 0.7250  | 1.1652  | 0.7563   | 0.029    | 0.117    | 0.540    | 0.229     |
> | IL-RL interwave        | 0.3181       | 0.6334      | 1.068       | 0.6733       | 0.117        | 0.171        | 0.537        | 0.275         |
> | two-stage              | 2.4303  | 4.2050  | 6.0318  | 4.2224   | 2.286    | 4.127    | 6.531    | 4.315     |
> | decouple + competitive | 0.2953  | 0.6004  | 1.0252  | 0.6403   | 0.088    | 0.127    | 0.433    | 0.216     |
>
> The loss-merging strategy underperforms because it mixes gradients from two objectives that are not always aligned. The two-stage approach, although usually used in LLM post-training, but here under our setting performs the worst. We believe this is due to distribution misalignment: the world model, having been pretrained solely on expert demonstrations via IL, fails to generalize to the novel state distributions encountered during RL-driven exploration.
>
> In contrast, our decouple + competitive approach achieves the best balance. It allows each policy to specialize and evolve independently while selectively exchanging knowledge through latent-space rollouts and reward-based selection. This setup avoids direct objective interference and produces more robust planning behavior, validating the motivation behind our dual-policy design.
>
> # Q2: Policy usage statistics
> Thank you for pointing this out. Below we provide the concrete cumulative win counts of the IL and RL actors throughout training, where a “win” means the policy achieved a higher reward.
> | Iteration | IL Wins | RL Wins | IL Win Rate |
> |-----------|---------|---------|-------------|
>  | 10k | 84 | 5 | 94.38% |
>  | 20k | 180 | 7 | 96.26% |
>  | 30k | 266 | 13 | 95.34% |
>  | 40k | 335 | 28 | 92.29% |
>  | 50k | 416 | 40 | 91.23% |
>  | 60k | 500 | 40 | 92.59% |
>  | 70k | 543 | 57 | 90.50% |
>  | 80k | 574 | 70 | 89.13% |
>
> As the table shows, the IL actor dominates the RL actor in most comparisons (with win rate consistently above 89%). In the early phase of training (before 60k iterations), imitation from expert demonstrations allows the IL actor to outperform RL consistently. However, as training progresses, the RL actor—leveraging knowledge gained from IL—begins to explore more effectively, leading to a gradual increase in its win rate.

---

> > ### Comment · Reviewer_yZWB · 2025-08-03
> >
> > **Response to W1**
> > As an RL practitioner, I remain unconvinced that the proposed method would be broadly beneficial beyond your presented setting. In the current setting, the dual-policy approach shows only marginal improvements over pure IL, with no evidence that it would help in other settings. To make this appealing to the RL community and convincingly show that the mechanism helps (which I believe is beyond the scope of this work, and why I see it as more suitable for an AD-focused community), you should compare a **standard SOTA IL method** (possibly also offline RL such as IQL) against a **SOTA RL method** (e.g., Dreamer, TD-MPC2) and against your dual-policy approach on standard benchmarks, using more seeds, so that the performance gain or loss can be clearly measured.
> >
> > **Core contribution vs. evidence**
> > In the paper (L212), you note that there is no improvement in L2 error over SSR but a significant improvement in collision rate.
> > However, in the ablations you report:
> >
> > * w/o competitive IL Actor: 0.249 avg. collision rate
> > * w/ competitive IL Actor: 0.216 avg. collision rate
> >   But in (Q1) you report pure-IL: 0.222 avg. collision rate.
> >
> > This inconsistency suggests that you should increase the number of samples to obtain more reliable comparisons. In the current setting, gains over pure IL appear marginal. This does not convincingly isolate the benefit of the dual-policy mechanism. Please report aggregated results across tasks with confidence intervals and number of seeds. You do not report running multiple seeds — in fact, in the reproducibility checklist you explicitly state that you did **not** run enough seeds to get statistical significance, but you “believe the experiment result is reproducible.”
> >
> > After reviewing the ablations, it appears that in this setting one could simply use IL and avoid the added complexity of the dual-policy framework.
> >
> > **Conclusion**
> > Given the small margins over IL-only, the high IL usage rate, and the lack of multiple seeds or significance testing, I am not yet convinced that the dual-policy mechanism is the source of meaningful benefit. My recommendation remains.

---

> > > ### Author Response · Authors · 2025-08-06
> > > **Response to Reviewer yZWB's valuable feedback (part 1)**
> > >
> > > Thank you very much for your detailed and thoughtful feedback. We truly appreciate the opportunity to communicate with you.
> > >
> > > We address your two main concerns below.
> > > ## Regarding the result inconsistency
> > > > there is no improvement in L2 error over SSR but a significant improvement in collision rate. However, in the ablations, gains over pure IL appear marginal.
> > >
> > > You're absolutely right to point out the importance of clarity and consistency in our reported results. Let us explain the context:
> > >
> > > In our main paper, the backbone used is SSR, which is relatively weak (Col% avg = 0.44%), making it easy to show large improvements in collision rate. However, we later discovered that SSR was unstable — running it twice with the same seed yielded different results. For this reason, we switched to LAW during the rebuttal phase, as it provides deterministic results under the same seed and setup. While LAW’s baseline performance is stronger (Col% avg = 0.22%), this also makes it more challenging to show dramatic gains, hence the observed marginal improvements.
> > >
> > > > w/ competitive RL Actor: 4.930 avg. collision rate, and in (Q1) you report pure-RL: 4.930 avg. (two result are same)
> > >
> > > > w/ competitive IL Actor: 0.216 avg. collision rate But in (Q1) you report pure-IL: 0.222 avg. (two results are not the same).
> > >
> > > Yes, at the rebuttal phase, due to the time limit, we use the result of “w/ competitive RL Actor” as the result of “pure-RL” (we forgot to mention that, sorry). However, in Q1, “pure-IL” refers to the original LAW backbone without any dual-policy or competition. The “w/ competitive IL Actor” setup still includes the dual-policy mechanism. Apologies for the confusion caused by overlapping results — but the numbers themselves are consistent. We confirm that our current experimental setup yields fully reproducible results when using the same seed and device.
> > >
> > > > You do not report running multiple seeds — in fact, in the reproducibility checklist you explicitly state that you did not run enough seeds to get statistical significance, but you “believe the experiment result is reproducible.”
> > >
> > > You’re right — we did not run multiple seeds due to computational constraints (each run requires 20 hours on 8× A800-80G). Still, we absolutely agree with your point. We are currently running additional experiments with multiple seeds and will update the results in the comments as they complete.
> > >
> > > ## On generalization and broader applicability
> > >
> > > > In the current setting, the dual-policy approach shows only marginal improvements over pure IL, with no evidence that it would help in other settings
> > >
> > > We understand your concern. While the improvement may appear marginal in open-loop evaluation on LAW, we found our approach provides notable gains in generalization and long-tail scenarios. Please see the results below:
> > >
> > > **Cross-city Generalization (Train on Singapore → Eval on Boston)**
> > >
> > > |                     | L2 (1s) | L2 (2s) | L2 (3s) | L2 (avg) | Col (1s) | Col (2s) | Col (3s) | Col (avg) |
> > > |---------------------|---------|---------|---------|-----------|-----------|-----------|-----------|------------|
> > > | LAW                 | 0.60997 | 1.20124 | 1.90667 | 1.23929   | 0.133     | 0.923     | 2.220     | 1.092      |
> > > | our method (LAW)    | 0.38826 | 0.78194 | 1.32027 | 0.83016   | 0.133     | 0.209     | 0.539     | 0.294      |
> > >
> > > **Long-tail Dataset Comparison (L2-based selection)**
> > > |                    | L2 (1s) | L2 (2s) | L2 (3s) | L2 (avg) | Col (1s) | Col (2s) | Col (3s) | Col (avg) |
> > > |--------------------|---------|---------|---------|-----------|-----------|-----------|-----------|------------|
> > > | LAW                | 0.3753  | 0.7388  | 1.2117  | 0.7753    | 0.0000    | 0.0795    | 0.4590    | 0.1795     |
> > > | our method (LAW)   | 0.3172  | 0.6518  | 1.1143  | 0.6944    | 0.0000    | 0.0794    | 0.3531    | 0.1441     |
> > >
> > > **Long-tail Dataset Comparison (Collision-based selection)**
> > > |                    | L2 (1s) | L2 (2s) | L2 (3s) | L2 (avg) | Col (1s) | Col (2s) | Col (3s) | Col (avg) |
> > > |--------------------|---------|---------|---------|-----------|-----------|-----------|-----------|------------|
> > > | LAW                | 0.3360  | 0.6692  | 1.1837  | 0.7296    | 0.0000    | 0.8523    | 4.3561    | 1.7361     |
> > > | our method (LAW)   | 0.2996  | 0.6410  | 1.1126  | 0.6844    | 0.0000    | 0.2841    | 3.2197    | 1.1679     |
> > >
> > > These results suggest that our method does provide tangible benefits in more difficult and realistic situations — especially those not covered well by imitation learning alone.

---

> > > > ### Author Response · Authors · 2025-08-06
> > > > **Response to Reviewer yZWB's valuable feedback (part 2)**
> > > >
> > > > ## On your excellent suggestions
> > > >
> > > > You recommended comparing against:
> > > > - compare with **standard SOTA IL method (possibly also offline RL such as IQL)**,
> > > > - compare with **SOTA RL method (e.g., Dreamer, TD-MPC2)**,
> > > > - evaluate on **standard benchmarks**.
> > > >
> > > > We really appreciate these ideas. However, to our best knowledge, pure RL methods are rarely applied directly on open-loop datasets like nuScenes/Navsim due to lack of interaction and sparse feedback. Most industry work trains in closed-loop simulators like CARLA, where many recent works (e.g., Think2Drive) adopt world models like DreamerV3. However, expert demonstration is lacking in these close loop simulators.
> > > >
> > > > It is intuitive to first use IL for pretraining and then apply RL for fine-tuning, leveraging both expert demonstrations and interactive exploration. This Real2Sim2Real paradigm has gained popularity in embodied AI and autonomous driving research and applications.
> > > >
> > > > In contrast, our goal in this paper is to explore whether simple RL signals can be integrated during the pretraining stage, alongside IL, even in open-loop settings. So far, we have shown that even lightweight RL supervision can lead to measurable improvements in performance.
> > > >
> > > > In future work, we plan to extend this direction by incorporating more powerful and standard RL algorithms, as you suggested.
> > > >
> > > > Thank you again for your constructive critique, insightful suggestions, and valuable time. Your feedback has been deeply appreciated and will meaningfully guide our future research.

---

> > > > ### Author Response · Authors · 2025-08-08
> > > > **Summary for Reviewer yZWB**
> > > >
> > > > We sincerely thank you for your detailed and insightful feedback throughout the discussion. Below is a summary of the key concerns you raised and how we’ve addressed them:
> > > >
> > > > 1. **Reproducibility and Result Consistency**
> > > >
> > > >     **Concern:** Inconsistent results between ablations and main table; lack of multiple seeds; small margins over IL.
> > > >
> > > >     **Our response:**
> > > >
> > > >     - We clarified that the earlier SSR backbone had reproducibility issues, so we switched to the stronger and stable LAW backbone for rebuttal.
> > > >
> > > >     - We explained the source of overlap in reported numbers (e.g., “pure-RL” reused “w/ competitive RL” result).
> > > >
> > > >     - Although limited by computing resources (each run takes ~20h on 8×A800), we will run multiple seeds and update the results in our manuscript. And we guarantee that using the same seed and same device, our results are reproducible.
> > > >
> > > > 2. **Generalization Beyond the Reported Setting**
> > > >
> > > >     **Concern:** Marginal gains in the current setting; unclear broader impact.
> > > >
> > > >     **Our response:**
> > > >
> > > >     - We presented additional results showing significant improvements in cross-city generalization and on long-tail scenarios.
> > > >
> > > >     - These gains suggest that while open-loop averages are marginal, the dual-policy setup improves robustness under harder and unseen conditions.
> > > >
> > > > 3. **Applicability to RL Community & Benchmark Comparisons**
> > > >
> > > >     **Concern:** Limited comparison to SOTA IL/RL (e.g., IQL, Dreamer, TD-MPC2), and limited use of standard benchmarks.
> > > >
> > > >     **Our response:**
> > > >
> > > >     - We explained that pure RL is rarely applied to open-loop datasets like nuScenes/Navsim due to non-interactivity.
> > > >
> > > >     - Our work explores integrating simple RL signals during the IL pretraining phase, even in non-interactive settings, which is a novel yet practical direction.
> > > >
> > > >     - We agree with your suggestion and plan to test our approach with more standard RL methods and across broader benchmarks in the future.
> > > >
> > > > We would love to hear any further thoughts you may have in the final hours, and thank you again for your expert input and engagement~

---

> > > > > ### Author Response · Authors · 2025-08-09
> > > > > **Update newest results**
> > > > >
> > > > > We’re excited to share our latest results:
> > > > >
> > > > > | Method     | L2 (1s) | L2 (2s) | L2 (3s) | L2 avg | Col% (1s) | Col% (2s) | Col% (3s) | Col% avg |
> > > > > |------------|---------|---------|---------|--------|-----------|-----------|-----------|----------|
> > > > > | LAW        | 0.3224  | 0.6274  | 1.0299  | 0.6599 | 0.088     | 0.122     | 0.456     | 0.222    |
> > > > > | Ours (LAW) | 0.3132  | 0.6105  | 1.0146  | 0.6461 | 0.039     | 0.078     | 0.479     | 0.199    |
> > > > >
> > > > > We sincerely thank you again for your time, constructive feedback, and guidance, your input has been instrumental in driving these improvements.

---

### Official Review · Reviewer_K5jx · 2025-07-08

**Clarity:** 2
**Significance:** 2
**Originality:** 3
**Rating:** 4
**Confidence:** 5

**Summary:**

This paper presents a hybrid approach that combines imitation learning (IL) and reinforcement learning (RL) for end-to-end autonomous driving. Given the challenges faced by existing end-to-end methods such as generalization,  the method introduces a computationally efficient method by combining  IL and RL in latent space. In particular, it introduces a competitive dual-policy framework  where IL learns from expert demonstrations via supervised latent rollouts, while the other (RL) explores the same latent environment using Dreamer-style imagined rollouts. Rather than jointly optimizing a single policy, the two agents train independently and compete: after each iteration, the better performer “blends” its knowledge into the other, enabling both to specialize and benefit from one another. The paper performs experiments on nuScenes and Navsim benchmarks. It shows improved performance in terms of collision rate.

**Questions:**

I have listed all my concerns in the "Strength and Weaknesses section". Please refer to "Weaknesses" for the raised concerns and questions.

**Ethical Concerns:**

["NO or VERY MINOR ethics concerns only"]

**Final Justification:**

I would like to thank the author for their responses. I think the paper has merits, but lacked a lot of fundamental experiments.  While, rebuttal have addressed some of the concerns,  performance improvement remain fluctuating and marginal and performance still way lower than the counterpart VLM methods. Based on the author responses in rebuttal, it seems if they could do a diligent work in a longer period than rebuttal, they can have a much stronger paper. Moreover, I think the author should position this paper in a different position compared to VLMs. They should do a through fair comparison study in inference using same computational resources. I would like to have seen the VAD inference time compared to their work as it shares the same latency as VLM based methods and show how their methods can benefit end-to-end methods from such aspects (e.g. maybe this method can enable a much larger scene encoder which can provide further improvements while having better latency than counterpart methods).   While I am rating the paper as borderline accept due to diligent work in addressing some of the concerns in rebuttal, I would like to encourage authors to substantially improve the manuscript.

**Quality:**

2

**Strengths And Weaknesses:**

Strengths

1)	The paper introduces a novel strategy for end-to-end autonomous driving. In particular, leveraging a dual policy framework where joint training of imitation and reinforcement learning towards enhanced E2E driving is a promising direction

2)	The proposed approach operates on latent space which provides computational efficiency over counterpart methods that leverages reconstruction techniques.

3)	The proposed method shows improvement in terms of collision rate at expense of L2 error.

Weaknesses

1)	Introduction rightfully mentions the generalization problem of E2E models. However, recent E2E methods [1,2, etc] leverages VLMs in an computationally efficient way to address the same problem. The paper lacks mentioning such works and does not provide any comparison.

2)	In Section, 3.1. author argues “human often plan actions backward” and then formulate their problem accordingly. What are the analytical/empirical evidence (e.g. ablation study) to back such claims?

3)	The paper mentions generalization and long-tail issues, but there is no specific experiment showing how proposed method helps in such specific scenarios. Provided experiments are generic (e.g. collision rate, L2 error on the whole nuscenes dataset). a) How does the proposed method works when it is trained in one city (e.g. Boston-nuscenes) and tested in another one (e.g. Singapore-nuscenes). b) How would the proposed method work on challenging long-tail scenarios (see [2] for reference)

4)	The paper mentions the computational efficiency of the method, but fails to provide comparisons with existing methods in terms of computational parameters e.g.  FPS, number of params .

5)	The proposed method does not provide impressive performance, e.g. requires trade-off between L2 and Collision Rate. Other methods addressing same problem such as VLM methods provides improved performance across both metrics.

[1] Vlp: Vision language planning for autonomous driving, CVPR 2024 [2] Distilling Multi-modal Large Language Models for Autonomous Driving, CVPR 2025

---

> ### Author Rebuttal · Authors · 2025-07-31
>
> # Meta Summary
>
> We sincerely thank all reviewers for their thoughtful feedback. Below, we summarize the core concerns raised across reviews.
> **We hope this summary helps the reviewers and area chair better understand the key review points and how we address them in our detailed responses:**
>
> 1. **Motivation and Justification for Dual-Policy Design**
>    Reviewers questioned the rationale behind combining IL and RL, and whether simpler alternatives (e.g., IL-pretrained world models for RL) would suffice. Several reviewers (yzWB, UrXC) requested ablations to isolate IL and RL contributions and clarify the necessity of the dual-policy framework.
>
> 2. **Clarity and Mechanism of Policy Blending**
>    Reviewers (yzWB, FwhP, uMEo) noted limited explanation of the competitive learning mechanism, including the blending method (soft vs. hard), threshold selection, and how arbitration occurs during training and inference.
>
> 3. **Effectiveness of Competition and Knowledge Transfer**
>    Concerns were raised about the empirical and theoretical support for the proposed competition mechanism. Reviewers asked for clearer evidence of its benefit and stability during learning.
>
> 4. **Motivation for Inverse Autoregressive Planning**
>    The rationale for planning actions in reverse (inverse autoregression) was seen as counterintuitive and insufficiently justified. Reviewers (UrXC, K5jx) requested both technical motivation and ablation results.
>
> 5. **Generalization and Long-Tail Evaluation**
>    Reviewers (K5jx, FwhP) emphasized the importance of testing the method in rare or challenging scenarios and questioned whether the hand-designed reward components can generalize to complex real-world driving situations.
>
>
> # Q1: Lack related works of VLM-based method
> We appreciate the reviewer’s feedback and acknowledge the omission of recent VLM-based methods in our introduction and related work sections. We will revise the paper to include relevant VLM-based approaches, such as [1, 2, etc.], and discuss their connection to our work.
>
> However, we think that a direct comparison between our method and VLM-based approaches may not be fair. These methods typically leverage VLMs which are trained on massive corpora, which go well beyond the nuScenes dataset in terms of scale. Moreover, VLM-based approaches introduce an additional language modality, which fundamentally shifts the modeling paradigm.
>
> While we agree that some VLM-based methods have demonstrated computational efficiency at inference time, it is important to note that they often require costly supervision or training pipelines. And fine-tuning large VLM also remains nontrivial and resource-intensive.
>
> [1] Distilling Multi-modal Large Language Models for Autonomous Driving. CVPR 2025
>
> [2] VlP: Vision Language Planning for Autonomous Driving. CVPR 2024
>
> [3] AutoVLA: A Vision-Language-Action Model for End-to-End Autonomous Driving with Adaptive Reasoning and Reinforcement Fine-Tuning
>
> [4] OpenDriveVLA: Towards End-to-end Autonomous Driving with Large Vision Language Action Model
>
> [5] ORION: A Holistic End-to-End Autonomous Driving Framework by Vision-Language Instructed Action Generation
>
> # Q2: The confusing Inverse Autoregression usage
> In our method, the planning head generates trajectories in a backward manner. Our Motivation stems from the observations that humans often plan actions by reasoning backward from the goal. Imagine a man is sitting at a desk, and to his right is a water bottle, and an iPad is between his hand and the bottle. When the person feels thirsty and want to drink, the person will:
>
> 1. First attend to the water bottle as the goal;
> 2. Then realize the iPad is blocking direct access;
> 3. Finally, plan the action: raise hand, over ipad, and get the bottle.
>
> This example illustrates how backward planning works: starting from the goal and adjusting earlier steps accordingly. Actually, backward planning has been explored in the field of embodied AI. For instance, [6] adopted a similar backward modeling strategy and gained good planning performance and efficiency.
>
> Empirical Results: Before adopting Inv. AR, we conducted ablations comparing:
> - Baseline (no autoregression)
> - Baseline + Autoregression (AR)
> - Baseline + Inverse Autoregression (Inv. AR)
>
> The results are shown below:
>
> | Setting               | L2 (1s) | L2 (2s) | L2 (3s) | L2 (avg) | Col (1s) | Col (2s) | Col (3s) | Col (avg) |
> |-----------------------|--------|--------|--------|----------|----------|----------|----------|-----------|
> | Our model             | 0.2114 | 0.3932 | 0.6642 | 0.4227   | 0.1758  | 0.3272   | 0.4428   | 0.3153    |
> | Our model + AR        | 0.1943 | 0.3709 | 0.6429 | 0.4027   | 0.1660   | 0.2735   | 0.5600   | 0.3332    |
> | Our model + Inv. AR   | 0.1999 | 0.3773 | 0.6519 | 0.4097   | 0.09768   | 0.1660   | 0.3288   | 0.1975    |
>
> We found that after using the inverse autoregressive mechanism, L2 and Collision Rate got a better score. The experiment results push us to use Inv. AR in our model.
>
> [6] Efficient Robotic Policy Learning via Latent Space Backward Planning.
>
> # Q3: Lack evaluation results to test the model’s generalization ability and performance on long-tail scenarios
> Thanks for your suggestion. We have compared the generalization ability and performance on long-tail scenarios.
>
> As for the generalization ability test, we first train the model on nuScenes-sinapore, then eval it on nuScenes-boston.
>
> |                     | L2 (1s) | L2 (2s) | L2 (3s) | L2 (avg) | Col (1s) | Col (2s) | Col (3s) | Col (avg) |
> |---------------------|---------|---------|---------|-----------|-----------|-----------|-----------|------------|
> | LAW                 | 0.60997 | 1.20124 | 1.90667 | 1.23929   | 0.133     | 0.923     | 2.220     | 1.092      |
> | LAW + our method    | 0.38826 | 0.78194 | 1.32027 | 0.83016   | 0.133     | 0.209     | 0.539     | 0.294      |
>
> To evaluate long-tail scenes, we extract a long-tail subset from nuScenes. Specifically, the subset contains scenes with two conditions: high L2 trajectory error or high collision rate. We tested our model on the challenging long-tail subset and compared with the baseline mode. The results indicate that our model achieved stronger performance on the long-tail dataset.
> The test data is summarized in the table below:
>
> Long-tail dataset comparison (L2) results
>
> |                    | L2 (1s) | L2 (2s) | L2 (3s) | L2 (avg) | Col (1s) | Col (2s) | Col (3s) | Col (avg) |
> |--------------------|---------|---------|---------|-----------|-----------|-----------|-----------|------------|
> | LAW                | 0.3753  | 0.7388  | 1.2117  | 0.7753    | 0.0000    | 0.0795    | 0.4590    | 0.1795     |
> | LAW + our method   | 0.3172  | 0.6518  | 1.1143  | 0.6944    | 0.0000    | 0.0794    | 0.3531    | 0.1441     |
>
> Long-tail dataset comparison (Collision) results
>
> |                    | L2 (1s) | L2 (2s) | L2 (3s) | L2 (avg) | Col (1s) | Col (2s) | Col (3s) | Col (avg) |
> |--------------------|---------|---------|---------|-----------|-----------|-----------|-----------|------------|
> | LAW                | 0.3360  | 0.6692  | 1.1837  | 0.7296    | 0.0000    | 0.8523    | 4.3561    | 1.7361     |
> | LAW + our method   | 0.2996  | 0.6410  | 1.1126  | 0.6844    | 0.0000    | 0.2841    | 3.2197    | 1.1679     |
>
> # Q4: Why you said your method is “computational efficient”?
>
> In our paper, we claim that our model is computationally efficient, particularly in comparison with methods that rely heavily on world models or reinforcement learning (RL) to boost performance. Below, we clarify the basis for this comparison:
>
> - Compared to pixel-level world models (e.g., Drive-WM[8], Panacea[9]): These approaches use generative world models to predict future images and apply supervision at the pixel level, which is computationally expensive. In contrast, our model performs in the latent space, which significantly reduces computational overhead.
> - Compared to pure RL-based methods (e.g., Think2Drive[10]): While these methods train driving policies in simulation environments like CARLA, they suffer from long training times due to the difficulty of policy convergence without expert demonstrations and sparse rewards. Our approach avoids this inefficiency, making it more computationally efficient in practice.
> - Compared to hybrid IL+RL approaches (e.g., RAD[11]): Some recent methods integrate IL with RL but introduce additional complexity. For example, RAD requires full 3D reconstruction of over 1000+ scenarios before policy training, which is an expensive computation cost. In contrast, our method uses a simple rule-based reward function and does not require 3D reconstruction, making the pipeline significantly more lightweight.
>
> # Q5: About the trade-off between L2 and Collision Rate.
> Recap: Compared with the best-performing model that does not incorporate auxiliary tasks, our model achieves the following results on the nuScenes benchmark:
> - L2 Error: 0.43 m, compared to 0.38 m from SSR — a 13% increase.
> - Collision Rate: 0.15%, compared to 0.30% from LAW — a 50% reduction.
>
> While we observe a trade-off between L2 error and collision rate, we believe it is a reasonable outcome. Firstly, collision rate is a more critical metric compared to L2 error, as safety must take precedence. In the real-world, drivers may produce diverse trajectories in the same situation for safety. Thus, it may be overly strict to expect the model to exactly mimic expert behavior. Furthermore, this trade-off validates the design of our system, which encourages competition between the IL and RL actors: the IL actor focuses on imitating expert demonstrations (leading to lower L2 error) and the RL actor explores beyond the demonstrations to discover safer trajectories (leading to lower collision rate). In this sense, the improved safety at the cost of slight deviation from expert behavior is acceptable and aligned with real-world priorities from our perspective.

---

> ### Comment · Reviewer_K5jx · 2025-08-04
> **Feedback**
>
> I would like to thank authors for providing answers to my concerns. The response to Q2 and Q3 are convincing. Please incorporate them to the paper. However, some of the points are still not convincing:
> 1) Comparison with VLMs. Both methods are aiming to address the same problem and VLMs have shown good generalization property without sacrificing computational efficiency. In autonomous driving, what matters is inference time not the training time. Moreover, finetuning MLLMs through parameter efficient methods such as LoRA provides good enough results. Thus, in my view, comparisons needs to be made.
> 2) I still did not get a concrete answer in terms of computational efficiency. What is latency in inference time (e.g. FPS, see Table 2 in [2] Distilling Multi-modal Large Language Models for Autonomous Driving, CVPR 2025 ). A paper can not claim computational efficiency in such venues without providing numbers.
> 3) The performance trade-off between L2 and collision rate is not convincing enough. Is there a way to get both metrics outperforming the counterpart methods. It is hard to prefer the proposed method over other counterpart methods that provide significant performance improvement on both open-loop metrics at no additional expense of computation.

---

> > ### Author Response · Authors · 2025-08-06
> > **Response to Reviewer K5jx's valuable feedback**
> >
> > Thanks for your feedback! Below is our response to your remaining three concerns:
> >
> > **Comparison with VLM-based models**:
> >
> > Indeed, both VLM-based models and world model + RL-based approaches aim to improve autonomous driving systems. Your suggestion is very reasonable. We have now added performance comparisons with DiMA and VLP (as mentioned in your comment) to Table 1 (see excerpt below). We believe this will provide readers with a broader and more informative perspective. Moreover, we also expect that future VLM/VLA architectures will integrate more tightly with world models and RL methods — an exciting trend already reflected in recent works like:
> >
> > [1] DriveAgent-R1: Advancing VLM-based Autonomous Driving with Hybrid Thinking and Active Perception
> >
> > [2] WorldVLA: Towards Autoregressive Action World Model
> >
> > **Concrete inference latency information**:
> >
> > You're absolutely right — inference time is critical for real-world deployment in autonomous driving. We have added the FPS and latency (in ms) metrics for our method in the updated table, measured on a single NVIDIA A100-40G. For VLP and DiMA, since their code is not public, we cite the inference results reported in their papers. The results show that although our L2 and Collision Rate are slightly worse than VLP and DiMA, our method offers faster inference, which is promising for practical applications.
> >
> >
> > **Trade-off between L2 and collision rate**:
> >
> >  We agree that an ideal method should improve both L2 error and collision rate. Our current RL modeling approach is relatively simple: we represent six waypoints using independent Gaussian distributions and sample directly from them. This often produces less smooth, non-human-like trajectories, making it difficult to reduce the L2 error. Similar challenges have been observed in prior work that combines IL and RL for autonomous driving, such as RAD [3], where introducing RL reduced the collision rate but slightly degraded human-likeness metrics like longitudinal jerk (see Table 1, CR and Long. Jerk columns).
> >
> > During the rebuttal phase, we replaced the weaker SSR backbone with LAW, a more robust ans stronger baseline (col% avg 0.22% v.s. 0.48% of SSR). While improvements are now more challenging to achieve, we observe gains in both L2 and collision rate, albeit modest ones. We plan to continue improving the RL policy by adopting more expressive modeling techniques to produce smoother and more accurate trajectories.
> >
> > [3] RAD: Training an End-to-End Driving Policy via Large-Scale 3DGS-based Reinforcement Learning
> >
> > | method     | L2 (1s) | L2 (2s) | L2 (3s) | L2 avg | Col% (1s) | Col% (2s) | Col% (3s) | Col% avg | fps   | latency (ms) |
> > |------------|---------|---------|---------|--------|-----------|-----------|-----------|-----------|--------|----------------|
> > | LAW        | 0.3224  | 0.6274  | 1.0299  | 0.6599 | 0.088     | 0.122     | 0.456     | 0.222     | 26.99 | 37.0           |
> > | Ours (LAW) | 0.2953  | 0.6004  | 1.0252  | 0.6403 | 0.088     | 0.127     | 0.433     | 0.216     | 27.1  | 36.97          |
> > | DiMA       | 0.18    | 0.36    | 0.61    | 0.38   | 0.07      | 0.10      | 0.27      | 0.15      | 16.8  | 59.5           |
> > | VLP-VAD    | 0.30    | 0.53    | 0.84    | 0.55   | 0.01      | 0.07      | 0.38      | 0.15      | -     | -              |

---

> > > ### Author Response · Authors · 2025-08-08
> > > **Summary for Reviewer K5jx**
> > >
> > > We sincerely thank Reviewer K5jx for their engagement and for acknowledging the responses to Q2 and Q3 as convincing.
> > > Remaining concerns and our responses:
> > > - **Comparison with VLMs (Q1):** We added VLP and DiMA results in our main table and discussed future trends of combining VLMs with world models and RL (e.g., DriveAgent-R1, WorldVLA).
> > >
> > >
> > > - **Inference latency (Q4):** We reported FPS and latency of our method (measured on A100-40G) and compared them with reported numbers from VLP/DiMA. Our method achieves faster inference speed.
> > >
> > >
> > > - **L2 vs. Collision Rate Trade-off (Q5):** We clarified that the RL trajectory modeling (six Gaussians) contributes to suboptimal L2. Despite this, when switching from the SSR to the LAW backbone, our method improves both L2 and collision rate, although modestly. We plan to adopt more natural trajectory modeling to enhance both metrics.
> > >
> > >
> > > We are happy to discuss further if you have any remaining concerns! Thank you again for your feedback and time~

---

> ### Author Response · Authors · 2025-08-09
> **Update the newest results**
>
> We’re excited to share our latest results — now **both L2 and collision rate show improvements** over the baseline!
>
> | Method     | LLM usage | L2 (1s) | L2 (2s) | L2 (3s) | L2 avg | Col% (1s) | Col% (2s) | Col% (3s) | Col% avg | FPS   | Latency (ms) |
> |------------|---------|---------|---------|---------|--------|-----------|-----------|-----------|----------|-------|--------------|
> | LAW        | no |0.3224  | 0.6274  | 1.0299  | 0.6599 | 0.088     | 0.122     | 0.456     | 0.222    | 26.99 | 37.0         |
> | Ours (LAW) | no | 0.3132  | 0.6105  | 1.0146  | 0.6461 | 0.039     | 0.078     | 0.479     | 0.199    | 27.10 | 36.97        |
> | DiMA       | yes |0.1800  | 0.3600  | 0.6100  | 0.3800 | 0.070     | 0.100     | 0.270     | 0.150    | 16.80 | 59.5         |
> | VLP-VAD    | yes | 0.3000  | 0.5300  | 0.8400  | 0.5500 | 0.010     | 0.070     | 0.380     | 0.150    |  -    |  -           |
>
> We sincerely thank you again for your time, constructive feedback, and guidance, your input has been instrumental in driving these improvements.

---

### Note · Authors · 2025-08-12

**Author Final Remarks**

**Recap**
We investigate whether RL signals can already benefit open-loop, imitation-style pretraining in embodied tasks (autonomous driving) without heavy simulators. Our method decouples IL/RL into two actors sharing a latent world model and exchanging knowledge via lightweight competition, and introduces inverse autoregression (Inv. AR) for planning.

**Rebuttal updates**
We found SSR unstable and moved to the stronger, reproducible LAW backbone. We added:
(i) VLM/VLA comparisons (VLP, DiMA)
(ii) inference latency (FPS/ms)
(iii) competition/blending details and usage statistics
(iv) deeper ablations of dual policy and Inv. AR
(v) new generalization and long-tail experiments

**New results (LAW)**
Both metrics improved: Col% avg 0.222 → 0.199, L2 avg 0.6599 → 0.6461.
Cross-city (train SG → eval Boston): L2 avg 1.239 → 0.830, Col% avg 1.092 → 0.294.
Long-tail (hard cases): L2 avg 0.730 → 0.684, Col% avg 1.736 → 1.168.
These gains target areas where IL struggles most—safety-critical and out-of-distribution scenarios.

**Addressing concerns**
• **Marginal IL gains** (yZWB, UrXC): On a stronger baseline, we reduce collision rate by 10% and achieve meaningful safety/generalization gains.
• **Breadth/generalization** (yZWB, UrXC): Evaluations on nuScenes/Navsim with SSR+LAW/WoTE plus zero-shot and long-tail analyses show broad utility in autonomous driving.
• **Cross-city & long-tail performance** (K5jx, FwhP): Collision rates drop 73% in zero-shot and 33% in hard long-tail cases.
• **Inv. AR feasibility** (UrXC): Provided ablations in our setting, examples from other fields, and psychological evidence.
• **IL dominance in competition** (yZWB, FwhP): IL wins ~90%, RL ~10%, but RL wins increase over time and still reduce collisions.

**Commitments for the revised paper**
We will include all new results, ablations isolating dual-policy/competition/Inv. AR on LAW, latency tables, VLM/VLA comparisons, and expanded Inv. AR motivation.

**Thanks**
We thank the reviewers and AC for their valuable guidance. Our work presents a practical, compute-light way to inject exploration during IL pretraining. Moving forward, we aim to investigate whether models trained with our method also gain in post-RL training, and to leverage the trained reward model for selecting the best trajectory proposal at inference, as inspired by Reviewer uMEo.

---

### Decision · Program_Chairs · 2025-09-17

**Decision:**

Reject

**Comment:**

The paper proposes a methods that combines IL and RL. The IL agent and the RL agent are share a common latent world model, but they are trained independently and compete during learning. Competition results in selecting which data should be shared between agents, either the data collected by the exploratory agent or the expert demonstrations. The goal is to allow policy specialization while agents benefiting from each other’s strengths. Experiments on two environments/datasets illustrate the benefits of their approach.

Reviewers expressed concerns about the evaluation, especially around missing VLM baselines with SOTA performance and lack of evidence of generalization claims and real benefits of the proposed approach; as well as poor motivation of the design choices, especially the benefit of the RL component and the trade-off between L2 placement error and collision rate.
The authors provided extensive rebuttal, running more experiments, including VLM baselines, measuring inferente time (to motivate against VLMs), ablations, etc. Although reviewers appreciated the new results, they also found them a bit rushed and pointed to new concerns, like potential unfair comparisons on inference time by not considering the same computational resources, or some reproducibility issues that the authors seemed to fix by changing the backbone.

My recommendation is reject. Even though the paper introduces a novel and interesting framework (different from the standard approach of adding objective functions) and even though the authors showed promising results during the rebuttal phase, I believe the changes are substantial, and a more systematic review of the paper would make it much stronger.